# Mixing Mechanisms: How Language Models Retrieve Bound Entities In-Context

**Yoav Gur-Arieh**[♠◇]**, Mor Geva**[♠]**, Atticus Geiger**[◇♡]

[♠]Blavatnik School of Computer Science and AI, Tel Aviv University
[◇]Pr(Ai)$^2$R Group
[♡]Goodfire

## ABSTRACT

A key component of in-context reasoning is the ability of language models (LMs) to bind entities for later retrieval. For example, an LM might represent *Ann loves pie* by binding *Ann* to *pie*, allowing it to later retrieve *Ann* when asked *Who loves pie?* Prior research on short lists of bound entities found strong evidence that LMs implement such retrieval via a **positional mechanism**, where *Ann* is retrieved based on its position in context. In this work, we find that this mechanism generalizes poorly to more complex settings; as the number of bound entities in context increases, the positional mechanism becomes noisy and unreliable in middle positions. To compensate for this, we find that LMs supplement the positional mechanism with a **lexical mechanism** (retrieving *Ann* using its bound counterpart *pie*) and a **reflexive mechanism** (retrieving *Ann* through a direct pointer). Through extensive experiments on nine models and ten binding tasks, we uncover a consistent pattern in how LMs mix these mechanisms to drive model behavior. We leverage these insights to develop a causal model combining all three mechanisms that estimates next token distributions with 95% agreement. Finally, we show that our model generalizes to substantially longer inputs of open-ended text interleaved with entity groups, further demonstrating the robustness of our findings in more natural settings. Overall, our study establishes a more complete picture of how LMs bind and retrieve entities in-context.

## 1    INTRODUCTION

Language models (LMs) are known for their ability to perform in-context reasoning (Brown et al., 2020), and fundamental to this capability is the task of connecting related entities in a text—known as *binding*—to construct a representation of context that can be queried for next token prediction. However, LMs are also known for struggling in reasoning tasks over long contexts (Liu et al., 2024; Levy et al., 2024). In this work, we conduct a mechanistic investigation into the internals of LMs to better understand how they bind entities in increasingly complex settings.

Neural networks' ability to bind arbitrary entities was a central issue in connectionist models of cognition (Touretzky & Minton, 1985; Fodor & Pylyshyn, 1988; Smolensky, 1990) and has reemerged in the era of LMs as a target phenomenon for mechanistic interpretability research (Davies et al., 2023; Prakash et al., 2024; 2025; Feng & Steinhardt, 2024; Feng et al., 2024; Wu et al., 2025). For example, to represent the text *Pete loves jam and Ann loves pie*, an LM will bind *Pete* to *jam* and *Ann* to *pie*. This enables the LM to answer questions like *Who loves pie?* by querying the bound entities to retrieve the answer (*Ann*). The prevailing view is that LMs retrieve bound entities using a **positional mechanism** (Dai et al., 2024; Prakash et al., 2024; 2025), where the query entity (*pie*) is used to determine the in-context position of *Ann loves pie*—in this case, the second clause after *Pete loves jam*—which is dereferenced to retrieve the answer *Ann*.

In this work, we show that position-based retrieval holds only for simple settings. This mechanism is unreliable for the middle positions in long lists of entity groups—a pattern that echoes the "lost-in-the-middle" effect (Liu et al., 2024) in LMs as well as primacy and recency biases in both humans (Ebbinghaus, 1913; Miller & Campbell, 1959) and LMs (Janik, 2024). To compensate for this noise,

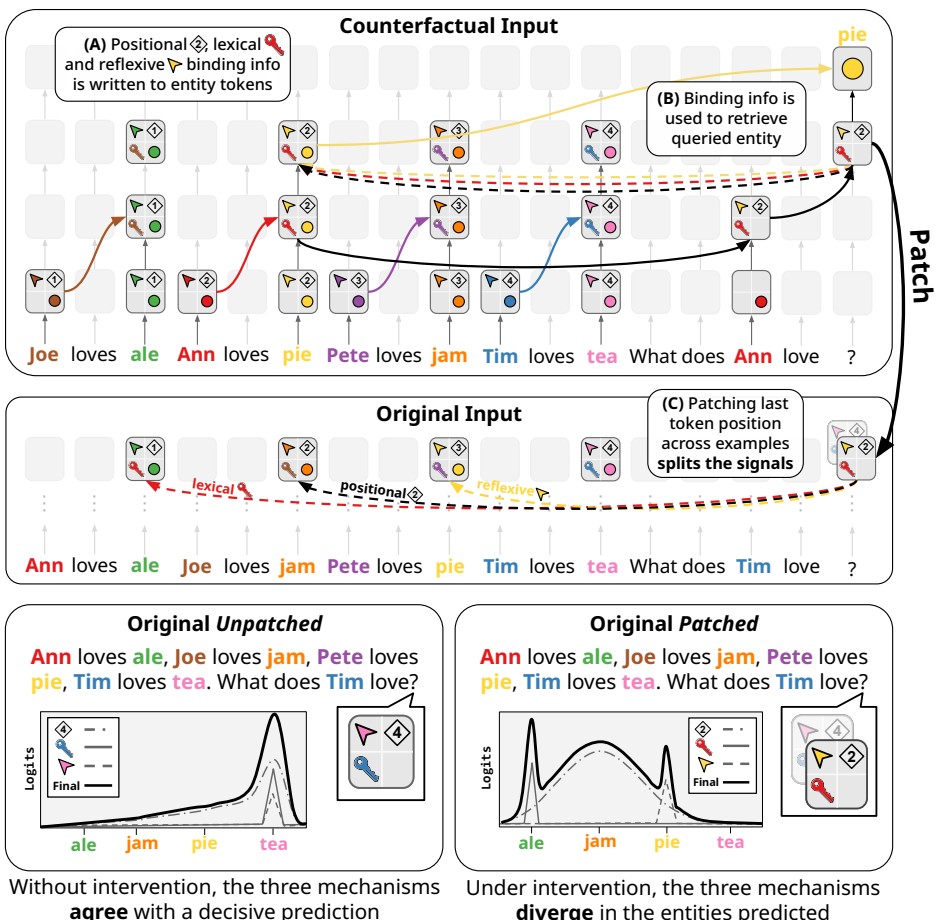

Figure 1: An illustration of the three mechanisms for retrieving bound entities in-context. We find that as models process inputs with groups of entities: (A) binding information of three types—positional, lexical, reflexive—is encoded in the entity tokens of each group, (B) this binding information is jointly used to retrieve entities in-context, and (C) it is possible to separate the three binding signals with counterfactual patching. The counterfactual input is designed such that patching activations to the LM run on the original input results in the positional, lexical, and reflexive mechanisms predicting different entities (See §3.2). The lexical signal from the counterfactual picks out *ale* in the original, because the question in the counterfactual was about *Ann*. The positional signal from the counterfactual picks out *jam*, because the question in the counterfactual was about the second character. The reflexive signal picks out *pie*, because *pie* was the counterfactual answer.

LMs supplement the positional mechanism with a **lexical mechanism**, where the query entity (*pie*) is used to retrieve its bound counterpart (*Ann*), and a **reflexive mechanism**, where the queried entity (*Ann*) is retrieved with a direct pointer that was previously retrieved via the query entity (*pie*).

In a series of ablation experiments, we show that all three mechanisms are necessary to develop an accurate causal model of the next token distribution (Pîslar et al., 2025), and that their interplay depends on the positions of the query entity (*pie*) and the retrieved entity (*Ann*). This mixture of mechanisms is robustly present across (1) the Llama, Gemma, and Qwen model families, (2) model sizes within those families ranging from 2 to 72 billion parameters, and (3) ten variable binding tasks. By better understanding this mechanism, we take a step toward explaining both the strengths and the persistent fragilities of LLMs in long-context settings, as well as the fundamental mechanisms that support in-context reasoning. We release our code and data at `https://github.com/yoavgur/mixing-mechs`.

## 2 PROBLEM SETUP AND PRIOR WORK

**Entity Binding Tasks**   In our experiments, we design a number of templatic in-context reasoning tasks with a similar structure to the example from the introduction, i.e., *Pete loves jam, Ann loves pie. Who loves pie?* Formally, a task consists of:

1. **Entity Roles**: Disjoint sets of entities $\mathcal{E}_1, \ldots, \mathcal{E}_m$ that will fill particular roles in a templatic text. For example, the set $\mathcal{E}_1$ might be names of people $\{Ann, Pete, Tim, \ldots\}$, and the set $\mathcal{E}_2$ might be foods and drinks $\{ale, jam, pie, \ldots\}$.

2. **Entity Groups**: An entity group is a tuple $G \in \mathcal{E}_1 \times \cdots \times \mathcal{E}_m$ containing entities that will be placed within the same clause in a template. For example, we could set $G_1 = (Pete, jam)$ and $G_2 = (Ann, pie)$. For convenience, we define $\mathbf{G}$ as a binding matrix wherein $\mathbf{G}_i^j$ denotes the $j$-th entity in the $i$-th entity group.

3. A **template** ($\mathcal{T}$): A function that takes as input a binding matrix $\mathbf{G}$, the *query entity* $q = \mathbf{G}_{q_{\text{entity}}}^{q_{\text{group}}}$, and the target entity $t = \mathbf{G}_{t_{\text{entity}}}^{q_{\text{group}}}$. Here $q_{\text{group}}$ is a positional index of the entity group containing the target and query, and $t_{\text{entity}} \neq q_{\text{entity}}$ index the positions of the target and query entities within that group, respectively. See §A.1 for more details and examples.

Continuing our example, define

$$\mathcal{T}(\mathbf{G}, q, t) = G_1^1 \text{ loves } G_1^2, G_2^1 \text{ loves } G_2^2. \begin{cases} \text{Who loves } q? & t_{\text{entity}} == 1 \\ \text{What does } q \text{ love?} & t_{\text{entity}} == 2 \end{cases}$$

and observe that

$$\mathcal{T}\left(\begin{bmatrix} Pete & jam \\ Ann & pie \end{bmatrix}, pie, Ann\right) = \textit{Pete loves jam, Ann loves pie. Who loves pie?}$$

For our experiments, the binding matrix $\mathbf{G}$ will consist of distinct entities.

**Interchange Interventions**   To probe the mechanisms an LM uses to bind and retrieve entities, we employ interchange interventions (Vig et al., 2020; Geiger et al., 2020; Finlayson et al., 2021; Geiger et al., 2021), the standard tool for prior work on binding and retrieval (Davies et al., 2023; Feng & Steinhardt, 2024; Prakash et al., 2024; 2025; Wu et al., 2025). These interventions allow us to identify which hidden states are causally relevant for the model in entity binding, by running the LM on paired examples — an *original input* and a *counterfactual input* — and replacing selected components, e.g., residual stream vectors, in the original run with those from the counterfactual.

**Causal Abstraction**   We develop a causal model of LM internals (Geiger et al., 2021; 2025b;a) that predicts the LM next token distribution using a mixture of three mechanisms (See §4). To test our hypotheses, we construct a dataset of paired originals and counterfactuals such that an interchange intervention on the causal model results in the positional, lexical and reflexive mechanisms increasing the probability of distinct tokens. To evaluate our proposed causal model and various ablations, we perform interchange interventions on the causal model and the LM, measuring the similarity between the next token distribution of the two models, and average across a dataset.

**Prior Studies of Entity Binding in LMs**   Previous work paints a picture of how entity binding and retrieval is performed by LMs. First, LMs bind together a group of entities by aggregating information about all entities in the entity token at the last position in the group. By co-locating information about entities in the residual stream of a single token, the LM can later on use attention to retrieve information about one bound entity conditional on a second bound entity (Feng & Steinhardt, 2024; Feng et al., 2024; Dai et al., 2024; Prakash et al., 2024; Wu et al., 2025), an algorithmic motif that Prakash et al. (2025) dub a "lookback" mechanism. We study the "pointers" used in the lookback mechanism that bring the next token prediction into the residual stream of the final token. We include experiments on the "addresses" contained in the residual streams of the bound entity tokens, as well as the query token, in §C.

Prior works identify a positional mechanism that is utilized in entity binding (described in detail in §3.1), but either evaluate it only in narrow settings (Prakash et al., 2025) or achieve low causal

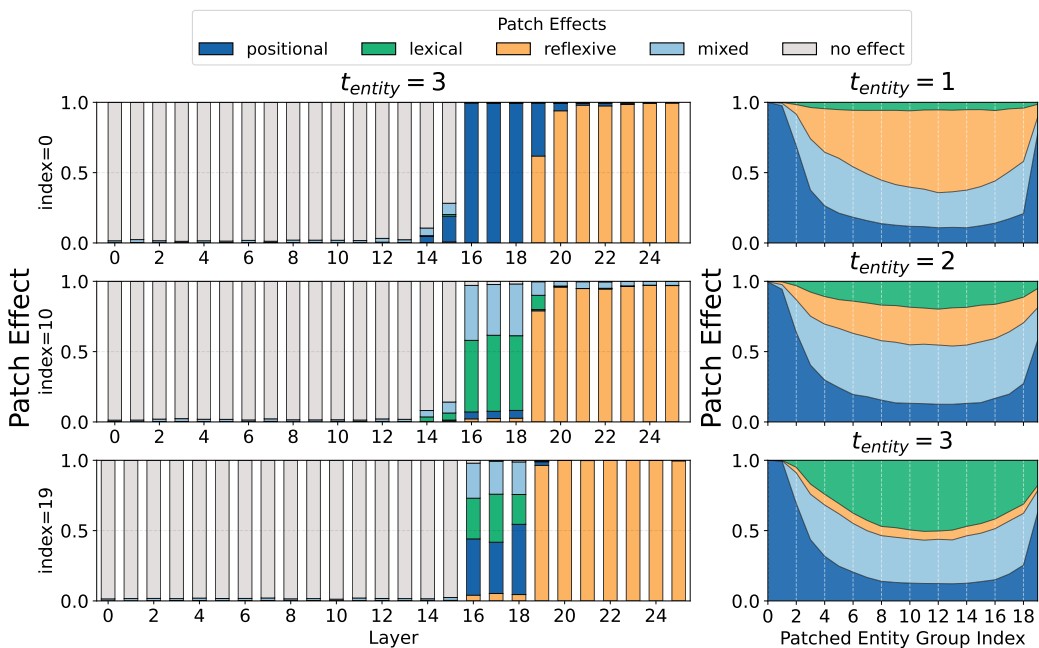

Figure 2: Results from interchange interventions on gemma-2-2b-it over a counterfactual dataset with three entities per group ($m = 3$) (See Figure 1 and §3.2). Outputs predicted by the positional, lexical and reflexive mechanisms are shown in dark blue, green and orange. In light blue, we show the cases not predicted by any of the mechanisms, dubbed *mixed*. These cases are further analyzed in §3.3. **Left**: Distribution of effects (y-axis) for three representative entity group indices (first, middle, and last) with $t_{\text{entity}} = 3$ for all layers (x-axis). At layers 16–18, the last token position carries binding information used for retrieval. **Right**: Distribution of effects (y-axis) for all entity indices (x-axis) at layer 18 for $t_{\text{entity}} \in \{1, 2, 3\}$, i.e., the question can be about any of the three entities in each clause. A U-shaped curve emerges: first and last indices rely more on the positional mechanism, while middle indices rely more on the lexical and reflexive mechanisms. See §A.2 for replication across models and tasks, and Figure 20 for plots using the original prompt as the x-axis.

faithfulness in predicting model behavior solely using this mechanism (Prakash et al., 2024; Dai et al., 2024). Feng & Steinhardt (2024); Prakash et al. (2025) restrict their analysis to queries of the final token in a group ($t_{\text{entity}} = m$) and to very small contexts ($n \in 2, 3$). Prakash et al. (2024) and Dai et al. (2024) find a positional mechanism in longer contexts ($n = 7$), but with low faithfulness.

# 3 THREE MECHANISMS FOR RETRIEVING BOUND ENTITIES

In this section, we define the positional mechanism and propose two alternative mechanisms (§3.1), all three of which make distinct claims about the causal structure of the LM. Then, we design a dataset with pairs of original and counterfactual inputs, such that each of the three mechanisms makes distinct predictions under an interchange intervention with the pair (§3.2). Last, we perform interchange interventions on the full last-token residual stream vector at different layers of the LM and visualize the results so we can observe the interplay between the three mechanisms in the counterfactual behavior of the LM (§3.3). We detail in §3.3 and §D.2 how we localize the layers for conducting the interventions. In our experiments, we evaluate nine models—gemma-2-{2b/9b/27b}-it, qwen2.5-{3b/7b/32b/72b}-it, and llama-3.1-{8b/70b}-it—on two binding tasks, *boxes* and *music* (see Appendix Table 1). For gemma-2-2b-it and qwen2.5-7b-it, we evaluate on all ten binding tasks.

## 3.1 THE POSITIONAL MECHANISM AND TWO ALTERNATIVES

The prevailing view is that bound entities are retrieved with a positional mechanism, but we propose two alternatives: lexical and reflexive mechanisms. The positional, lexical, and reflexive mecha-

nisms are represented as causal models $\mathcal{P}$, $\mathcal{L}$, and $\mathcal{R}$ that each have single intermediate variables $P$, $L$, and $R$, respectively, used to retrieve an entity from context as the output.

**The Positional Mechanism**   Prior work provides evidence that a positional mechanism is used to retrieve an entity from a group via the group's positional index (Dai et al., 2024; Prakash et al., 2024; 2025). The model $\mathcal{P}$ indexes the group containing the query entity ($P := q_{\text{group}}$), and its output mechanism retrieves the target entity from the group at index $q_{\text{group}}$. In Figure 1, we have $P = 4$ when no intervention is performed on the LM and the target entity *tea* is retrieved from position 4, but after the intervention $P \leftarrow 2$ the entity *jam* at the second position is retrieved.

Although existing evidence shows that the positional mechanism explains LM behavior in settings with two or three entity groups (Prakash et al., 2025), it does not generalize. When more groups are introduced, the evidence is weaker (Prakash et al., 2024; Dai et al., 2024). Our goal is to investigate the failure modes of the positional mechanism as more entity groups are introduced, and to that end we propose two alternative hypotheses for how LMs implement binding.

**The Lexical Mechanism**   The *lexical* mechanism is perhaps the most intuitive solution: output the bound entity from the group containing the queried entity. The causal model $\mathcal{L}$ stores the query entity ($L := q$) and the output mechanism retrieves the target entity from the group containing $q$. In Figure 1, we have $L = $ *Tim* when no intervention is performed on the LM and the output mechanism retrieves the entity *tea* from the group with *Tim*. However, after the intervention $L \leftarrow Ann$, the entity *ale* is retrieved from the group with *Ann*.

**The Reflexive Mechanism**   The reflexive mechanism retrieves an entity with a direct, self-referential pointer—originating from that entity and pointing back to it (illustrated in Appendix Figure 7). However, if this signal is patched into a context where the token is not present, the mechanism fails. The model $\mathcal{R}$ stores the target entity ($R := t$) and the output mechanism retrieves the entity $t$ if it appears in context. In Figure 1, we have $R = $ *tea* when no intervention is performed and the entity *tea* is retrieved, but after the intervention $R \leftarrow pie$, the entity *pie* is retrieved because it appears in the original input.

The reflexive mechanism is an unintuitive solution, until one considers that the architecture of an autoregressive LM allows attention to only look from right to left. When the query occurs after a target in an entity group, i.e., $t_{\text{entity}} < q_{\text{entity}}$, the lexical mechanism is not possible. In the text *Tim loves tea*, the entity *tea* cannot be copied backwards to the residual stream of *Tim* so that the lexical mechanism can answer *Who loves tea?* Therefore, an earlier mechanism in the LM must first retrieve an absolute pointer that is in turn used to retrieve the bound entity token.

## 3.2   DESIGNING COUNTERFACTUAL INPUTS TO DISTINGUISH THE THREE MECHANISMS

We designed a dataset of paired original and counterfactual inputs such that the positional, lexical, and reflexive mechanisms will each make distinct predictions when an interchange intervention is performed on their respective intermediate variables, $P$, $L$, and $R$.

**Counterfactual Design**   Figure 1 displays a pair of original and counterfactual inputs that distinguish our three mechanisms (further detailed in Appendix Table 1). We illustrate this with the following example. Define the original and counterfactual binding matrices $\mathbf{G}$ and $\mathbf{G}'$ respectively:

$$\mathbf{G} = \begin{bmatrix} \text{Ann} & \text{ale} \\ \text{Joe} & \text{jam} \\ \text{Pete} & \text{pie} \\ \text{Tim} & \text{tea} \end{bmatrix} \qquad \mathbf{G}' = \begin{bmatrix} \text{Joe} & \text{ale} \\ \text{Ann} & \text{pie} \\ \text{Pete} & \text{jam} \\ \text{Tim} & \text{tea} \end{bmatrix} \tag{1}$$

We can then use the template $\mathcal{T}$ from §2 such that for these binding matrices, $\mathcal{T}(\mathbf{G}, \textit{Tim}, \textit{tea})$ yields the original input in Figure 1 and $\mathcal{T}(\mathbf{G}', \textit{Ann}, \textit{pie})$ yields the counterfactual input. Each of the three mechanisms produces a different output after an interchange intervention on this pair of inputs:

1.  An interchange intervention on $P$ in $\mathcal{P}$ would output the entity at the counterfactual query's position. Since $q'_{\text{group}} = 2$ for *Ann* in $\mathbf{G}'$, this sets $P \leftarrow 2$, and the output becomes *jam*.

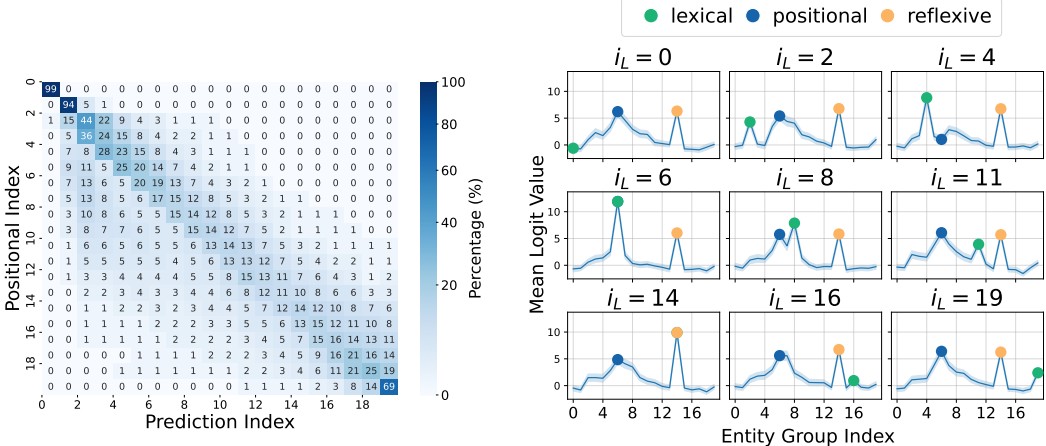

Figure 3: The positional mechanism is diffuse for middle entity groups. **Left**: Confusion matrix (%) of the patched positional index (y-axis) vs. gemma-2-2b-it's prediction (x-axis) after an interchange intervention (as in Figure 1). Counterfactual predictions cluster near the position promoted by the positional mechanism, decaying with distance. Only the *mixed* and positional patch effects from Figure 2 are shown; see Figure 31 for other models and tasks. **Right**: Mean logit distributions with $i_P = 6, i_R = 14$, and $i_L$ varied, illustrating additive and suppressive interaction between the three mechanisms. The lexical and reflexive signals form one-hot peaks, while the positional is broader and more diffuse. See Figures 24, 25, and 26 for more distributions.

2. An interchange intervention on $L$ in $\mathcal{L}$ would output the entity in the original input bound to the query entity in the counterfactual input. Since the query entity is now *Ann*, the mechanism queries the group containing *Ann* in **G** and outputs the bound entity *ale*.

3. An interchange intervention on $R$ in $\mathcal{R}$ would follow the direct pointer established in the counterfactual input. In this case, the pointer is to the token *pie*, which exists in the original input, and so the mechanism outputs *pie*.

Each of these three outputs is distinct from the actual output *tea* for the original input, which means the dataset also distinguishes the three mechanisms from no intervention being performed. Let $i_P$, $i_L$, and $i_R$ be indices of the entity groups queried by the positional, lexical, and reflexive mechanisms, e.g., $i_P = 2$, $i_L = 1$, and $i_R = 3$ in Figure 1 after patching. In our counterfactual datasets, each of the three mechanisms can predict any position in the list of entity groups from the original input, i.e., $i_P, i_L$, and $i_R$ vary freely from 1 to $n$. For details and task templates, see §A.1. A relevant remaining confounder is that the reflexive mechanism predicts the output that is the target entity in the counterfactual input, meaning this dataset cannot distinguish the pointer used by the reflexive mechanism from the actual next token prediction. We resolve this issue, validating the existence of a reflexive mechanism, in §3.4.

### 3.3 INTERVENTION EXPERIMENTS

We find experimentally that information used to retrieve a bound entity is accumulated in the last token residual stream across a subset of layers. In Figure 2, we show the results of interchange interventions on gemma-2-2b-it across the layers of the last token residual stream. We see that in layers 16–18 the model accumulates binding information in the last token position. Therefore, unless stated otherwise, we conduct all of our interchange interventions by patching the last token residual stream vector on the last layer before retrieval starts, denoted as $\ell$, which is different for each of the nine models we test, but consistent across tasks for a given model (see §D.2 for more details). We measure the next token distribution produced by the model under intervention and compare it against the possible outputs for the three mechanisms. We aggregate and visualize the results of these intervention experiments in Figures 2 and 3.

**The positional mechanism weakens for middle positions.** We can see plainly in Figure 2 that the positional mechanism controls behavior solely when the positional index is at the beginning or end of the sequence of $n = 20$ entity groups. In middle entity groups, however, its effect becomes minimal, accounting only for 20% of the model's behavior. Further analysis of the cases not explained by any of the mechanisms—dubbed *mixed* in the plot—reveals that these predictions are distributed near the positional index (Figure 3). Additionally, when collecting the mean logit distributions across many samples and fixing the positional index, we see that in the first and last positional indices it induces a strong and concentrated distribution around that index. However, in middle indices we see this distribution become wide and diffuse (Appendix Figure 13). Thus, the positional mechanism becomes unreliable in middle indices and cannot be used as the sole mechanism for retrieval. We show in §A.3 how this effect emerges as $n$, i.e., the total number of entity groups in context, increases, and in §G we disambiguate the effect of increasing $n$ from that of increasing sequence length.

**The lexical and reflexive mechanisms are modulated based on target entity position.** Observe in Figure 2 that when the positional mechanism is unreliable for middle positions, the lexical and reflexive mechanisms come into play. However, which of these two alternate mechanisms contribute more depends on the location of the target entity within the entity group, denoted as $t_{\text{entity}}$. When the target is at the beginning of the group ($t_{\text{entity}} = 1$), the reflexive mechanism is used (as discussed in §3.1). When the target is at the end ($t_{\text{entity}} = 3$), the lexical mechanism is primarily used. Finally, when the target is in the middle ($t_{\text{entity}} = 2$), both mechanisms are used to differing extents.

**The three mechanisms have complex interplay.** We can see in Figure 3 the interplay between the three mechanisms when the positional and reflexive indices are fixed to $i_P = 6$ and $i_R = 14$ while the lexical index $i_L$ is iterated over a range of values. First, the logit distributions clearly reveal the contributions of each mechanism, with a distinct spike appearing at each index. These spikes, however, behave differently. In line with Figure 3, the positional index produces a wide, diffuse distribution, whereas the lexical and reflexive indices produce sharp, one-hot peaks. Next, we observe that the mechanisms interact through a pattern of *competitive synergy*, meaning that they both boost and suppress one another. When the lexical index is close to the positional index, the lexical contribution is amplified while the positional contribution is weakened; when they are farther apart, neither affects the other. In contrast, when the lexical index is close to the reflexive index, the lexical contribution is suppressed by the reflexive one. We design more datasets of original-counterfactual pairs to understand the bound entity tokens' residual streams, and analyze how binding information is encoded and propagates across token positions, in §C.

**Takeaways** These results clarify how LMs bind and retrieve entities in context. They simultaneously employ three mechanisms: positional, lexical, and reflexive. In the first and last entity groups, LMs can rely almost exclusively on the positional mechanism, where it is strongest and most concentrated. In middle groups, however, the positional signal becomes diffuse and often links entities to nearby groups. In these cases, the lexical and reflexive mechanisms provide sharper signals which refine the positional mechanism, enabling the LM to retrieve the correct entity.

### 3.4 VALIDATING THE EXISTENCE OF THE REFLEXIVE MECHANISM

In our counterfactual design (§3.2), we constructed the counterfactuals such that each hypothesized mechanism makes a different prediction about the outcome of intervention experiments. However, we also noted that these counterfactuals fail to distinguish the pointer used in the reflexive mechanism from the answer itself. In this section, we address this by designing a new counterfactual dataset specific to distinguishing between the two. In §F we also conduct an attention knockout experiment to further strengthen our findings.

**New Counterfactual Design** We modify the existing dataset such that the counterfactual answer entity doesn't appear in the original input. For example, for the input pair from Figure 1, we would keep the original input *Ann loves ale, Joe loves jam, Pete loves pie, and Tim loves tea. What does Ann love?*, but alter the counterfactual to *Joe loves ale, Ann loves **cod**, Pete loves jam, and Tim loves tea. What does Ann love?* so the new counterfactual answer *cod* appears nowhere in the original.

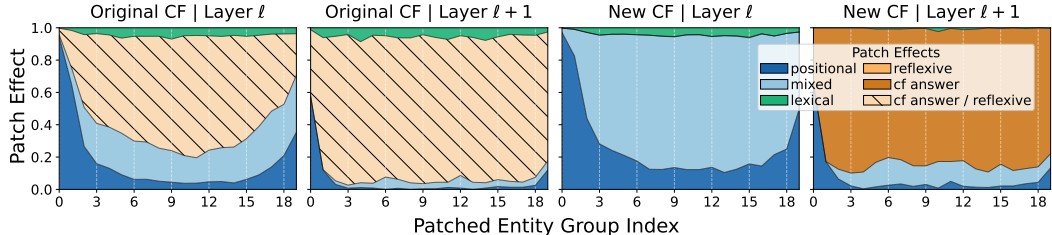

Figure 4: We distinguish the pointer in the reflexive mechanism from the answer entity with interchange interventions (gemma-2-2b-it, $t_{\text{entity}} = 1$, layers $\ell, \ell + 1$). **Left**: interventions on the original counterfactual dataset show that we can't distinguish between patching the pointer or the answer entity itself. **Right:** interventions on the modified counterfactuals (§3.4). At layer $\ell$ the model does not respond with the counterfactual answer entity which does not appear in the original context, indicating that the patched signal is a reflexive pointer that cannot be dereferenced. At layer $\ell + 1$, once the model has already retrieved the answer entity (§D.2), the patched signal becomes the answer entity itself. This shows that no confounding suppressive mechanism exists to prevent the model from answering with an entity not in its context.

Such examples differentiate between the pointer $R$ in the reflexive mechanism $\mathcal{R}$ and the output of the mechanism itself. An interchange intervention on the output of the mechanism would simply replace the answer entity *ale* with the answer entity *cod*. However, an interchange intervention on the pointer $R$ would patch in a pointer to the token *cod* that the mechanism $\mathcal{R}$ would attempt to dereference. However, *cod* does not appear in the original input, and thus the pointer cannot be resolved and no output is predicted by this mechanism.

**Results**  We show the results for layer $\ell$, for the original counterfactual setup, as well as for the new one, in Figure 4. We see that while in the original counterfactual setup, the model answered with the entity pointed to by the reflexive mechanism, under the new counterfactual setup it did not. This indicates that what was copied is a reflexive pointer that cannot be dereferenced, as opposed to the answer entity itself. One alternative explanation is that the model might contain a mechanism that suppresses outputs corresponding to entities absent from the context. To exclude this possibility, we repeat the evaluations at layer $\ell + 1$, a point at which the model has already retrieved the correct answer. Here, patching leads the model to output the counterfactual answer entity in both counterfactual setups, showing that no such suppressive mechanism is present. We can therefore conclude that the model indeed relies on a reflexive mechanism, distinct from the positional and lexical ones, where a direct pointer to the answer entity is used to retrieve it.

## 4 A SIMPLE MODEL FOR SIMULATING ENTITY RETRIEVAL IN-CONTEXT

To formalize our observations about the dynamics between the three mechanisms and the position of the target entity, we seek to develop a model that approximates LM logits for next token prediction, as a position-weighted mixture of terms for the positional, lexical, and reflexive mechanisms.

**Mixing mechanisms in a causal model**  We follow Pîslar et al. (2025) in combining together multiple causal models $(\mathcal{P}, \mathcal{L}, \mathcal{R})$ into a single causal model $\mathcal{M}$ that modulates between the mechanisms conditional on the input. In our combined causal model, the lexical and reflexive terms have separate learned weights conditioned on their respective index, i.e., $i_L$, or $i_R$. In accordance with the results shown in Figure 3, we model the lexical and reflexive mechanisms as one-hot distributions that up-weight only the target entity in groups $i_L$ and $i_R$, respectively. The positional term is modeled as a Gaussian distribution scaled by a single weight $w_{\text{pos}}$ centered at the index $i_P$ with a standard deviation that is a quadratic function of $i_P$. We define a new causal model $\mathcal{M}$ that uses all three variables $P$, $L$, and $R$ simultaneously to compute a logit value $Y_i$ for each entity $\mathbf{G}^i_{t_{\text{entity}}}$:

$$Y_i := \underbrace{w_{\text{pos}} \cdot \mathcal{N}\big(i \mid i_P, \sigma(i_P)^2\big)}_{\text{positional mechanism}} + \underbrace{w_{\text{lex}}[i_L] \cdot \mathbf{1}\{i = i_L\}}_{\text{lexical mechanism}} + \underbrace{w_{\text{ref}}[i_R] \cdot \mathbf{1}\{i = i_R\}}_{\text{reflexive mechanism}} \quad (2)$$

Where $\sigma(i_P) = \alpha(\frac{i_P}{n})^2 + \beta\frac{i_P}{n} + \gamma$. We learn $w_{pos}, w_{lex}, w_{ref}, \alpha, \beta, \gamma$ from data.

| Model | JSS ↑ | | |
|---|---|---|---|
| | $t_e = 1$ | $t_e = 2$ | $t_e = 3$ |
| *Comparing against the prevailing view* | | | |
| $\mathcal{M}\ (L_{\text{one-hot}}; R_{\text{one-hot}}; P_{\text{Gauss}})$ | **0.95** | **0.96** | **0.94** |
| $\mathcal{P}_{\text{one-hot}}$ (prevailing view) | 0.42 | 0.46 | 0.45 |
| *Modifying the positional mechanism* | | | |
| $\mathcal{M}$ w/ $P_{\text{oracle}}$ | 0.96 | 0.98 | 0.96 |
| $\mathcal{M}$ w/ $P_{\text{one-hot}}$ | 0.86 | 0.85 | 0.85 |
| *Ablating the three mechanisms* | | | |
| $\mathcal{M} \setminus \{P_{\text{Gauss}}\}$ | 0.67 | 0.68 | 0.67 |
| $\mathcal{M} \setminus \{L_{\text{one-hot}}\}$ | 0.94 | 0.91 | 0.75 |
| $\mathcal{M} \setminus \{R_{\text{one-hot}}\}$ | 0.69 | 0.87 | 0.92 |
| $\mathcal{M} \setminus \{R_{\text{one-hot}}, L_{\text{one-hot}}\}$ | 0.69 | 0.84 | 0.74 |
| $\mathcal{M} \setminus \{P_{\text{Gauss}}, R_{\text{one-hot}}\}$ | 0.12 | 0.27 | 0.48 |
| $\mathcal{M} \setminus \{P_{\text{Gauss}}, L_{\text{one-hot}}\}$ | 0.55 | 0.41 | 0.20 |
| Uniform | 0.44 | 0.57 | 0.49 |

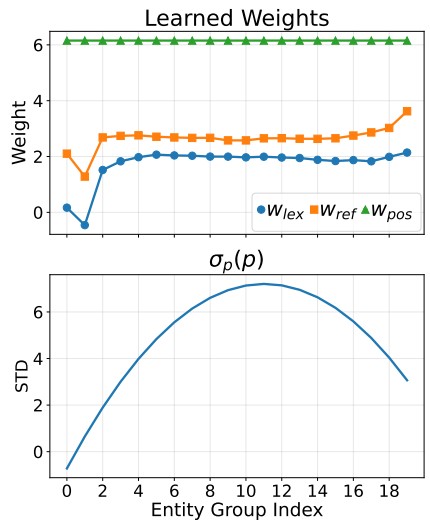

Figure 5: Results for training our full model $\mathcal{M}$ $(L_{\text{one-hot}}, R_{\text{one-hot}}, P_{\text{Gauss}})$, in addition to variants, baselines and ablations. **Left**: JSS scores for modeling the LM next token distribution over $i_P, i_L, i_R$. Evaluated on gemma-2-2b-it for the *music* binding task, with $t_e = t_{\text{entity}}$. Our model attains near-perfect JSS, slightly below the oracle. KL values (Table 3) show the same trend. All CIs are $< 0.02$; for $\mathcal{M}$ and $\mathcal{M}$ w/ oracle they are $< 0.002$. **Right**: Learned weights $w_{\text{lex}}, w_{\text{ref}}, w_{\text{pos}}$ and $\sigma$ curve, for $t_{\text{entity}} = 2$. Observe $\sigma$ widens for middle indices and narrows toward the end.

**Learning how the mechanisms are mixed**  To generate data for training our causal model we performed 150 interchange interventions per combination of $1 \leq i_P, i_L, i_R \leq n$ using the original and counterfactual inputs designed to distinguish the three mechanisms (see Figure 1 and Section 3.2). We collected the logit distributions per index combination, and averaged them into mean probability distributions by first applying a softmax over the entity group indices and then taking the mean. This yields $n^3 = 8,000$ probability distributions, which serve as our data for training and evaluation. We used 70% of the data for learning the causal model parameters and split the remainder evenly between validation and test sets. The loss used is the Jensen–Shannon divergence (JSD) between our model's predicted probability distribution and the target, chosen for its symmetry.

We evaluate $\mathcal{M}$ alongside a range of baselines, variants, and ablations to characterize our model's performance and understand the contribution of the different mechanisms. Experiments are run with gemma-2-2b-it on the *music* task ($n = 20$, $t_{\text{entity}} \in [3]$). In §E we report the same setup for this model as well as qwen2.5-7b-it on additional tasks, with similar trends. We measure similarity between the predicted and target distributions using Jensen–Shannon similarity (JSS), defined as $1 - \text{JSD}$, calculated with $\log_2$ to yield values in $[0, 1]$. See Appendix Table 3 for KL divergences.

We compare our model with: (1) The prevailing view – a one-hot distribution at the positional index, (2) a variant of $\mathcal{M}$ using a one-hot distribution at $i_P$ instead of a Gaussian, (3) ablations of $\mathcal{M}$ using only a subset of the mechanisms (e.g., $\mathcal{M} \setminus \{L_{\text{one-hot}}\}$ omits the lexical term from Equation 2), and (4) a uniform distribution. Finally, as an upper bound, we evaluate an oracle variant, where the lexical and reflexive components are learned as usual, but the positional component is swapped with the actual logit distributions of the model, as a function of $i_P$ (see Figure 13).

**Results**  Figure 5 shows the results. Our model achieves near-perfect performance, only slightly below the oracle, at an average JSS of 0.95. In contrast, the prevailing view achieves an average JSS of 0.44, well below even the uniform distribution baseline with 0.5. Next, we see that modeling the positional mechanism as a one-hot as opposed to a gaussian significantly hurts performance, dropping to 0.85 JSS. The ablations further reveal how mechanisms are employed: for instance, when $t_{\text{entity}} = 1$, ablating the lexical mechanism has nearly no effect, while for $t_{\text{entity}} = 3$ this is true for the reflexive mechanism. This is in keeping with previous results, showing that the lexical and

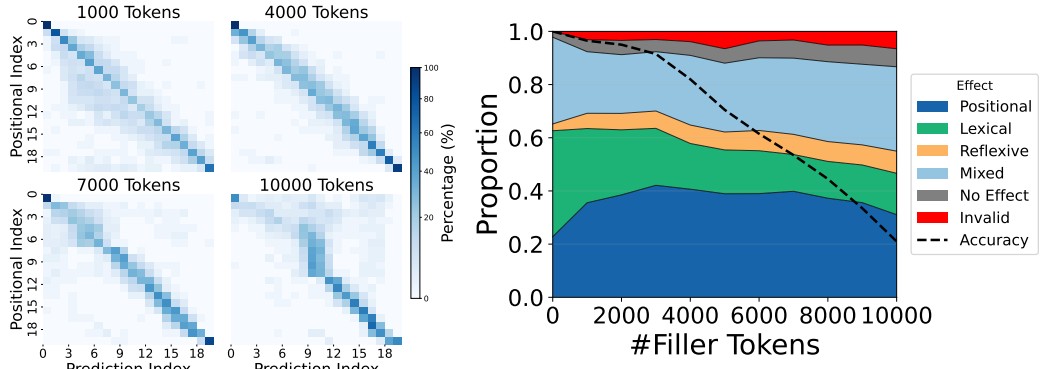

Figure 6: Padding results for gemma-2-2b-it on the *boxes* task. **Left**: Confusion matrix between the model's predicted index and the positional index patched in from the counterfactual. This gets increasingly fuzzy for early tokens as padding is increased. **Right**: Distribution of effects as padding is increased, showing the positional mechanism strengthens at the expense of the lexical mechanism.

reflexive mechanisms are used differently depending on the value of $t_{\text{entity}}$. Figure 5 also shows the learned parameters of the model for $t_{\text{entity}} = 2$. We see that in this setting, the lexical and reflexive mechanisms behave similarly—weaker at the beginning, flat in the middle, with an uptick at the end—although the reflexive mechanism is slightly more dominant, consistent with the table results. For the positional mechanism we can see that it starts off very concentrated, becoming wider in middle indices, and finally becoming more narrow towards the end, mirroring previous results.

## 5 INTRODUCING FREE FORM TEXT INTO THE TASK

To test our model's generalization to more realistic inputs, we modify our prompt templates $\mathcal{T}$ such that they include filler sentences between each entity group. To this end, we create 1,000 filler sentences that are "entity-less", meaning they do not contain sequences that signal the need to track or bind entities, e.g. "Ann loves ale, *this is a known fact*, Joe loves jam, *this logic is easy to follow*...". This enables us to evaluate entity binding in a more naturalistic setting, containing much more noise and longer sequences. We evaluate different levels of padding by interleaving the entity groups with an increasing number of filler sentences, for a maximum of 500 tokens between each entity group.

The results, shown in Figure 6 for gemma-2-2b-it on the *boxes* task, show that our model at first remains remarkably consistent in more naturalistic settings, across even a ten-fold increase in the number of tokens. However, as the amount of filler tokens increases, we see that the magnitude of the mechanisms' effects changes. The lexical mechanism declines in its effect, while the positional and mixed effects slightly increase. We can also see that the mixed effect remains distributed around the positional index, but that it slowly becomes more diffuse. Thus, when padding with 10,000 tokens, we get that other than the first entity group, the positional information becomes nearly non-existent for the first half of entity tokens, while remaining stronger in the latter half. This suggests that a weakening lexical mechanism relative to an increasingly noisy positional mechanism might be a mechanistic explanation of the "lost-in-the-middle" effect (Liu et al., 2024). In §D.4 we show that our model generalizes to inputs with more linguistic variability as well.

## 6 CONCLUSION

In this paper, we challenge the prevailing view that LMs retrieve bound entities purely with a positional mechanism. We find that while the positional mechanism is effective for entities introduced at the beginning or end of context, it becomes diffuse and unreliable in the middle. We show that in practice, LMs rely on a mixture of three mechanisms: positional, lexical, and reflexive. The lexical and reflexive mechanisms provide sharper signals that enable the model to correctly bind and retrieve entities throughout. We validate our findings across 9 models ranging from 2-72B, and 10 binding tasks, establishing a general account of how LMs retrieve bound entities.

## 7 REPRODUCIBILITY STATEMENT

We take multiple steps in this work to ensure the reproducibility of our findings. In §A.1 we detail all binding tasks used in our evaluations, and in §3.2 and Table 2 we describe how to construct datasets of paired original and counterfactual examples. §E specifies the hyperparameters used for training our causal models. The code for dataset generation and causal model training is included in the supplemental materials.

### ACKNOWLEDGMENTS

This work was supported in part by the Alon scholarship, the Israel Science Foundation grant 1083/24, and a grant from Open Philanthropy. Figure 1 uses images from `www.freepik.com`.

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

## A  EVALUATING GENERALIZATION

In this section, we seek to validate that our findings generalize to different values of $n$, as well as across models and binding tasks.

### A.1  BINDING TASKS

In this subsection, we detail the different binding tasks we evaluate, and show that our findings generalize across all of them. We define ten different binding tasks spanning domains, syntaxes, and subjects: one with $m = 2$ and nine with $m = 3$. The sizes of the entity sets range from 23 to 80. Table 1 lists the entity sets for each task, along with an example instantiation for $n = 2$ and different values of $q_{\text{entity}}$. Note that when $m = 3$, we use two query entities: $\mathbf{G}_{q_{\text{entity}}1}^{q_{\text{group}}}$ and $\mathbf{G}_{q_{\text{entity}}2}^{q_{\text{group}}}$. These are the two entities in the entity group which aren't the target entity. For example, when $t_{\text{entity}} = 2$ we set $q_{\text{entity}}1 = 1$ and $q_{\text{entity}}2 = 3$.

Figures 8 and 9 show the results of the `TargetRebind` interchange intervention on gemma-2-2b-it and qwen2.5-7b-it across all tasks and all values of $t_{\text{entity}}$. Our findings are consistent: the positional mechanism dominates for early and late entity groups, while the lexical and reflexive mechanisms take over in the middle. We also replicate the effect in Figure 2 and Figure 5: reflexive is more present when $t_{\text{entity}} = 1$ (first), lexical when $t_{\text{entity}} = 3$ (last), and both are balanced when $t_{\text{entity}} = 2$ (middle).

### A.2  REPLICATING RESULTS ACROSS MODELS

To validate robustness, we evaluated 9 models across 3 families, spanning 2–72B parameters. As shown in Figures 10 and 11 for the *boxes* and *music* tasks with $t_{\text{entity}} \in [3]$, our findings transfer consistently across models. The positional mechanism dominates for the first and last entity groups, while in middle positions lexical and reflexive take over, with a mixed effect distributed around the positional index. In the Qwen family, we also observe that positional efficacy strengthens with model size. Overall, these results point to a universal strategy used by LMs to solve entity binding tasks.

### A.3  EFFECT OF $n$

Previous work has described model behavior faithfully using only the positional mechanism (Feng & Steinhardt, 2024; Prakash et al., 2025), but these analyses were limited to small contexts ($n \in \{2, 3\}$). In this work, we show extensively that this finding doesn't hold for larger values of $n$. To evaluate exactly the relationship between the efficacy of the positional mechanism and $n$, we conduct the `TargetRebind` interchange intervention on gemma-2-2b-it and qwen2.5-7b-it for all $n \in [3, 20]$. We see in Figures 14, 15, 16, 17 that the trend seen in all experiments holds across values of $n$: the positional mechanism is effective in the first and last entity groups, but not in middle ones. Its efficacy for middle entity groups declines as $n$ increases. This trend is consistent with the separability analysis in Figure 22, which shows that hidden states from middle entity groups become increasingly difficult to classify by position as $n$ grows.

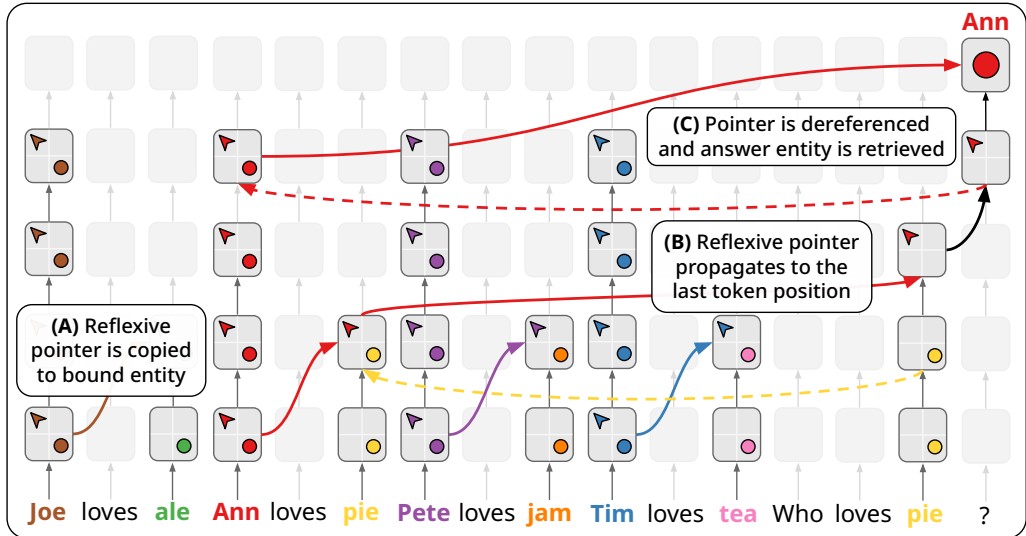

Figure 7: An illustration of the reflexive mechanism for retrieving entities for $t_{\text{entity}} = 1$. We omit the positional and lexical mechanisms for clarity. Under this mechanism: (A) a reflexive pointer to an entity originating from it is copied to its bound counterpart, (B) that reflexive pointer propagates to the last token position through the query entity, and (C) that pointer is dereferenced, thus retrieving the answer entity.

## B   ENCODING OF POSITIONAL INFORMATION

Throughout our experiments (notably Figures 2, 5, 10 and 8), we show that the model does not rely solely on the positional mechanism. One possible explanation is that, as illustrated in Figure 3, the positional signal becomes diffuse and weak for middle entity groups. This may reflect the model's limited ability to encode entity group positions in a linearly separable manner. To test this hypothesis, we collected hidden state activations at entity token positions as well as at final token positions at every layer, and assessed their separability using PCA and a multinomial logistic regression probe. Figure 22 shows the results: PCA projections for entity token positions with $n = 20$, and linear probe accuracies for both entity and final positions across $n \in \{5, 10, 15, 20\}$. Consistent with our broader findings, the first and last entity groups are readily separable, while middle groups exhibit substantial overlap. We also observe a clear dependence on $n$: smaller contexts yield better separation, whereas larger contexts make positions increasingly indistinguishable.

## C   BINDING SIGNALS IN ENTITY TOKENS

In our main experiments, we focus on interchange interventions for the last token position, showing that it encodes positional, lexical and reflexive signals. In this section, we conduct experiments to verify the existence of these signals in the entity token positions themselves, as well as identify the movement of these signals across token positions.

First, we conduct the `PosSwap`, `LexSwap` and `RefSwap` interchange interventions, described in Table 2, with the results shown in Figure 19. We see that they achieve nearly identical interchange intervention accuracies as when performing `TargetRebind` with the last token position. Additionally, we see that for the positional and lexical mechanisms, the crucial layers where the binding information is contained in the entity tokens and used for retrieval are layers 12-19, while for reflexive it's 6-12.

To further trace how binding signals flow through the model, we apply attention knockout (Geva et al., 2023). We first identify a minimal set of layers where blocking attention from the last token position to the query entity token (e.g., *which box is the **medicine** in?*) degrades performance. Across all values of $t_{\text{entity}}$, this occurs in layers 11–16, dropping accuracy from 98% to 37%, aligning with the layers where binding information resides in entity tokens. Knockout becomes even more

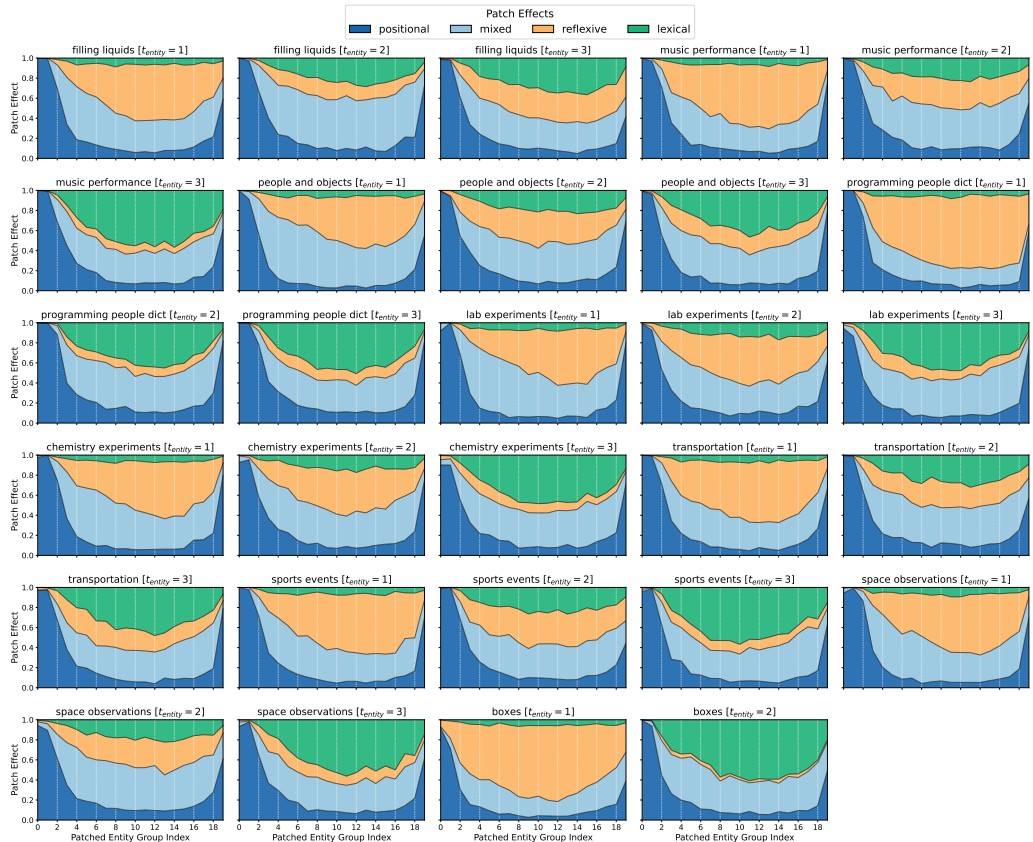

Figure 8: Results of the `TargetRebind` interchange intervention for gemma-2-2b-it across all tasks and possible values of $t_{\text{entity}}$.

effective when applied to both the query token and the token immediately after it, reducing accuracy to 8%. This suggests that some of the query signal is copied forward. Consistent with this, blocking attention from the last token position only to the token following the query token decreases accuracy by just one point. However, when we block attention both from the last position to the query token and from its following token, accuracy drops to 6%, confirming that crucial binding information reaches the last position via the query token.

Finally, we test whether binding information propagates from entity tokens to the query token. The lexical mechanism may not require such propagation, since its signal can be generated directly from the query token. By contrast, the reflexive signal in the entity tokens originates from the answer token, so the query token must retrieve it in order for the signal to reach the last token position. To evaluate this, we block attention from the query token to different entity tokens. For the reflexive signal (setting $t_{\text{entity}} = 1$), we block attention to the entity token identical to the query token—where our `RefSwap` intervention localized the signal—and to the token immediately after it. This intervention is most effective in layers 8–12, reducing accuracy to 6%, and matching the layers where entity tokens use this signal (Figure 19). Blocking attention to other entity tokens in the queried entity group has no effect. In contrast, for the lexical signal (setting $t_{\text{entity}} = 2$), blocking attention from the query token to the correct answer entity token reduces accuracy only to 86%, even when applied across all layers. Moreover, blocking attention from the query token to all entity tokens at all layers still leaves accuracy at 90%. These results support our hypothesis: the lexical signal can be derived locally from the query token, while the reflexive signal must be retrieved from entity tokens. This also explains why the model appears to produce the reflexive binding signal earlier than in the lexical or positional mechanisms – it requires an additional stage of retrieval.

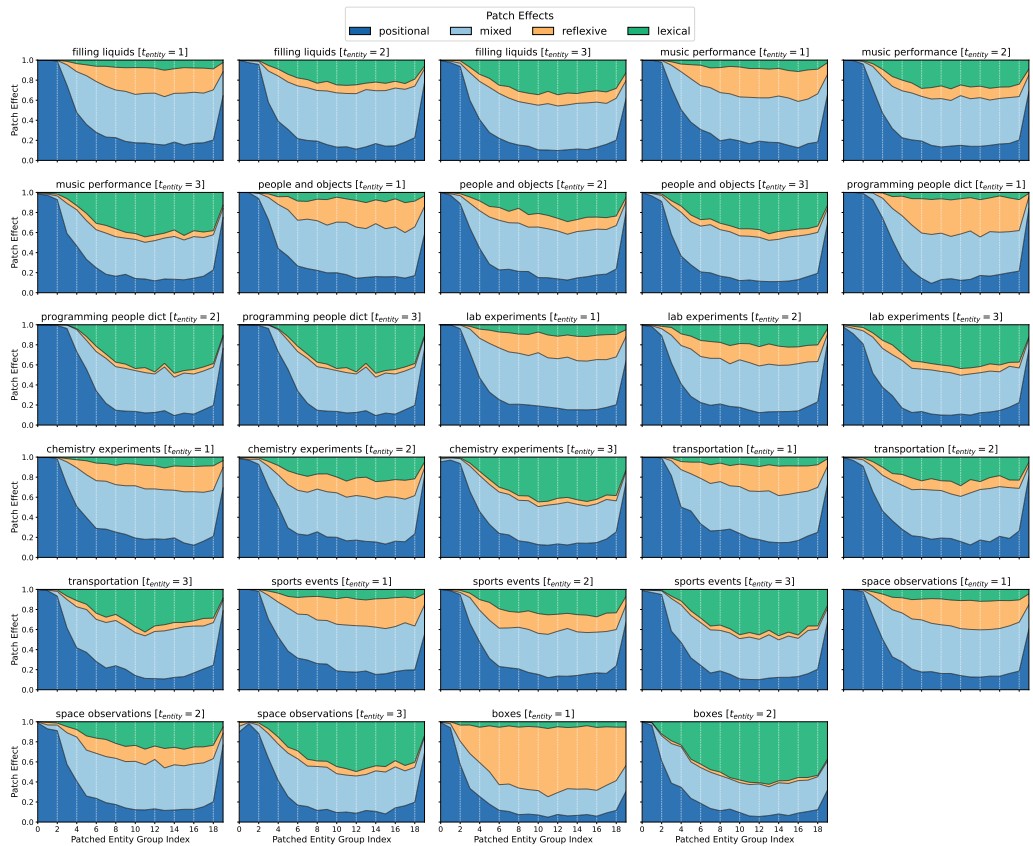

Figure 9: Results of the `TargetRebind` interchange intervention for qwen2.5-7b-it across all tasks and possible values of $t_{\text{entity}}$.

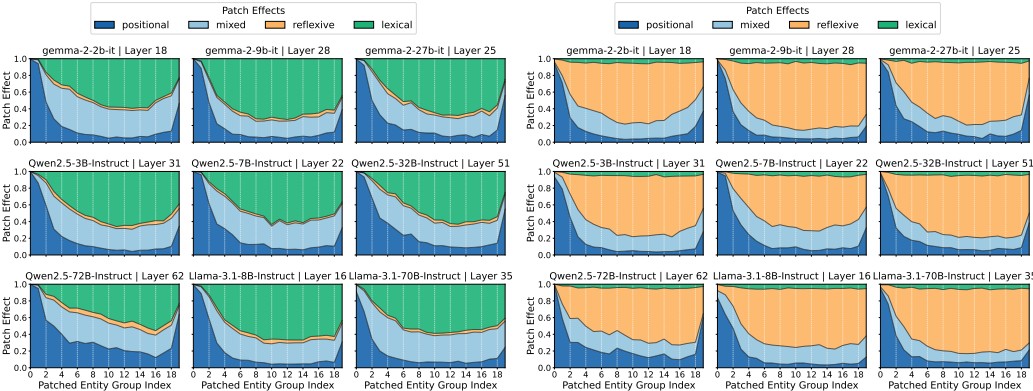

Figure 10: Evaluation of the `TargetRebind` interchange intervention in 9 different models across 3 model families spanning 2-72B parameters, for the *boxes* binding task and $t_{\text{entity}} \in [2]$. We see that the results remain remarkably consistent.

# D    ADDITIONAL EXPERIMENTS

In this section we discuss experiments that further strengthen our model of how LMs perform entity binding and retrieval, that couldn't be included in the main section. In §D.1 we expand our understanding of the interaction between different mechanisms by evaluating what happens when setting

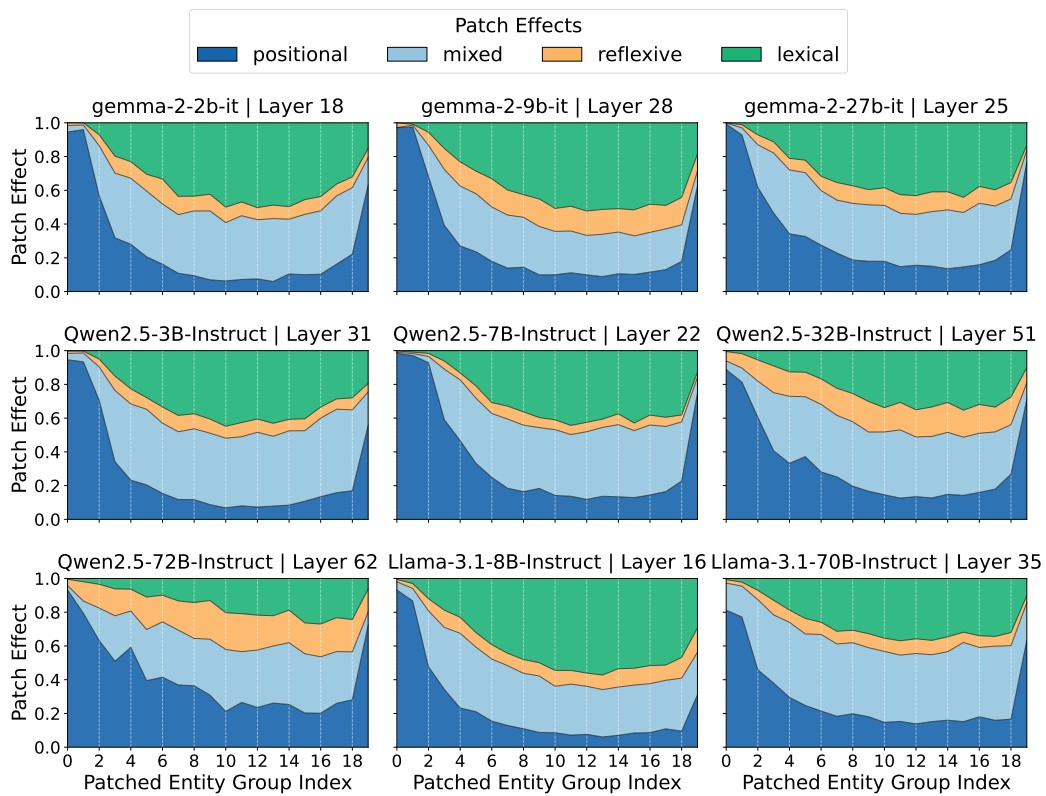

Figure 11: Evaluation of the `TargetRebind` interchange intervention in 9 different models across 3 model families spanning 2-72B parameters, for the *music* binding task and $t_{\text{entity}} = 3$.

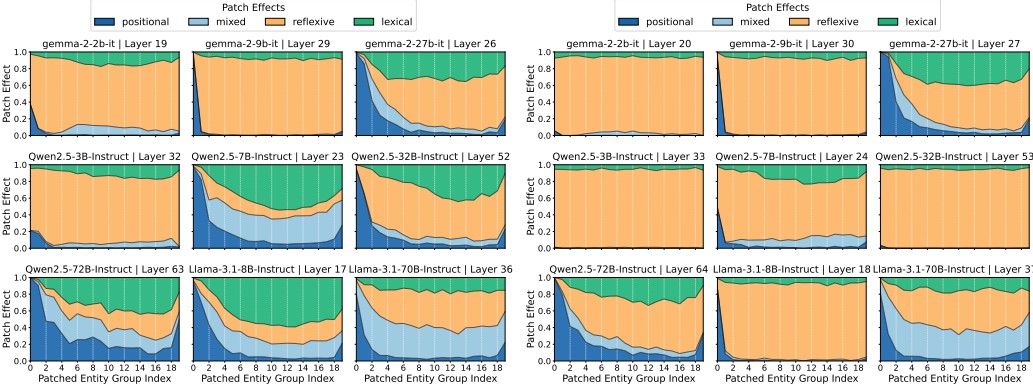

Figure 12: Evaluation of the `TargetRebind` interchange intervention at 1 and 2 layers after the evaluation in Figure 10, for $t_{\text{entity}} = 2$. We see that the model shifts from aggregating binding information to retrieving the entities.

two mechanisms to point at the same entity. In §D.2 we detail the experiments conducted for finding the target layer for our interchange interventions. Finally, in §D.3 we analyze the model's behavior when removing entities pointed to by the different mechanisms.

## D.1 MECHANISM AGREEMENT

In the `TargetRebind` interchange intervention used to produce the results in Figure 2 (and others throughout the paper), we explicitly make sure to have different values for the positional, lexical and

| Name | Entity Sets Sample | Binding Example |
|---|---|---|
| Filling Liquids | $\mathcal{E}_1 = \{John, Mary\}$
$\mathcal{E}_2 = \{cup, glass\}$
$\mathcal{E}_3 = \{wine, beer\}$ | John and Mary are working at a busy restaurant. To fulfill an order, John fills a cup with beer and Mary fills a glass with wine. Who filled a cup with beer? |
| People and Objects | $\mathcal{E}_1 = \{John, Mary\}$
$\mathcal{E}_2 = \{toy, medicine\}$
$\mathcal{E}_3 = \{kitchen, office\}$ | John put the medicine in the office and Mary put the toy in the kitchen. What did Mary put in the kitchen? |
| Programming Dictionary | $\mathcal{E}_1 = \{a, b\}$
$\mathcal{E}_2 = \{John, Mary\}$
$\mathcal{E}_3 = \{US, Canada\}$ | The following are dictionary variables in Python: a={'name':'Mary', 'Country':'Canada'}, b={'name':'John', 'Country':'US'}. What is the country in variable b where 'name' == 'John'? |
| Music | $\mathcal{E}_1 = \{John, Mary\}$
$\mathcal{E}_2 = \{rock, pop\}$
$\mathcal{E}_3 = \{guitar, piano\}$ | At the music festival, John performed rock music on the piano, and Mary performed pop music on the guitar. What music did Mary play on the guitar? |
| Biology Experiment | $\mathcal{E}_1 = \{John, Mary\}$
$\mathcal{E}_2 = \{serum, enzyme\}$
$\mathcal{E}_3 = \{beaker, vial\}$ | In a biology laboratory experiment, Mary placed the serum in a vial, and John placed the enzyme in a beaker. Who placed the serum in a vial? |
| Chemistry Experiment | $\mathcal{E}_1 = \{John, Mary\}$
$\mathcal{E}_2 = \{ethanol, acetone\}$
$\mathcal{E}_3 = \{crucible, funnel\}$ | In a chemistry laboratory experiment, Mary added the acetone to a crucible, and John added the ethanol to a funnel. What did John add to a funnel? |
| Transportation | $\mathcal{E}_1 = \{John, Mary\}$
$\mathcal{E}_2 = \{truck, taxi\}$
$\mathcal{E}_3 = \{mall, park\}$ | In a city transportation system, John drove the truck to the mall, and Mary drove the taxi to the park. Where did Mary drive the taxi? |
| Sports Events | $\mathcal{E}_1 = \{John, Mary\}$
$\mathcal{E}_2 = \{hockey, cricket\}$
$\mathcal{E}_3 = \{stadium, field\}$ | In a sports competition, Mary played hockey at the stadium, and John played cricket at the field. Who played hockey at the stadium? |
| Space Observations | $\mathcal{E}_1 = \{John, Mary\}$
$\mathcal{E}_2 = \{planet, asteroid\}$
$\mathcal{E}_3 = \{telescope, radar\}$ | During an astronomy study, John observed an asteroid with a radar, and Mary observed a planet with a telescope. What did John observe with a radar? |
| Boxes | $\mathcal{E}_1 = \{toy, medicine\}$
$\mathcal{E}_2 = \{box A, box B\}$ | The toy is in box B, and the medicine is in Box A. Which box is the medicine in? |

Table 1: List of all binding tasks we evaluate in our experiments. We show entity sets composed of only two entities per set for brevity. We also only show examples for $n = 2$ but evaluate over $n \in [3, 20]$

reflexive indices, so that we can know which mechanism most affected the model's output. However, as shown in Figure 3, these mechanisms behave additively, and we suspect that when they agree, they overwhelmingly drive model behavior. To evaluate this, we conduct two experiments, one for $t_{\text{entity}} = 1$ where the positional mechanism agrees with the reflexive one, and one for $t_{\text{entity}} = 3$ where the positional mechanism agrees with the lexical one. The results, shown in Figure 18, show that this is indeed the case.

## D.2 FINDING THE TARGET LAYER

We seek to identify for each model what the last layer before retrieval is, so that we can perform our interchange interventions on that layer. Indeed in Figure 2 we see that there are a subset of layers where the last token position contains the binding information, after which it contains the retrieved answer. Thus, for each model we identify the last layer where patching the last token position does not copy the retrieved token. The intervention on this layer $\ell$ is shown in Figure 10 for $t_{\text{entity}} \in [2]$, and in Figure 12 we show this same intervention for $\ell + 1$ and $\ell + 2$ with $t_{\text{entity}} = 2$. We see clearly that for $\ell$, the percentage of cases where the answer post-intervention is the retrieved entity from

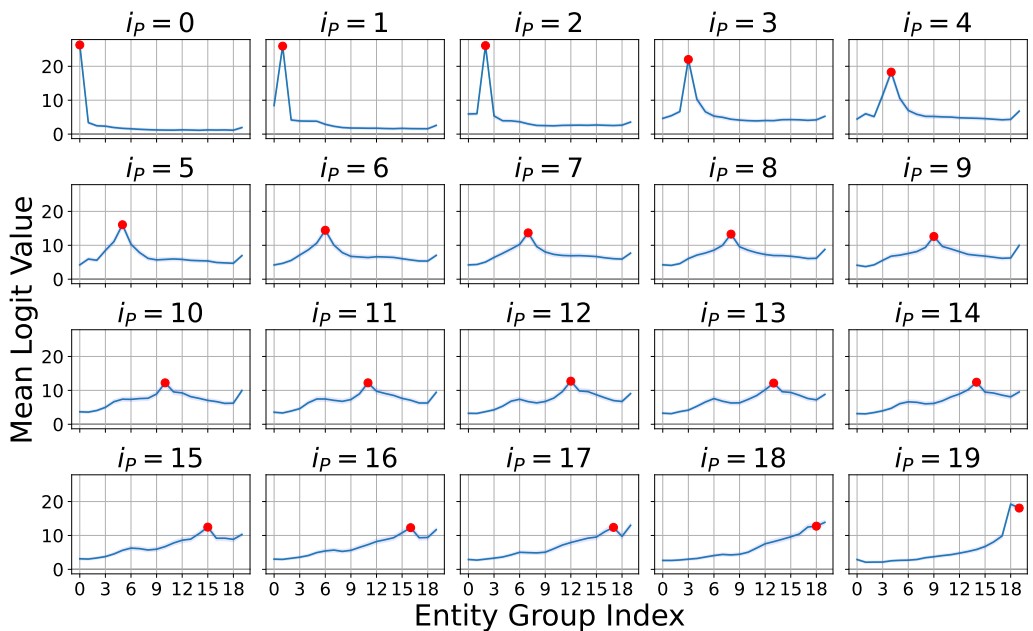

Figure 13: The mean logit distribution as a function of the positional index ($i_P$), for qwen2.5-7b-it on the *boxes* task with $t_{\text{entity}} = 2$. We can see the positional binding signal induces a strong and concentrated signal for entity groups in the beginning and the end, while inducing a weak and diffuse one for middle groups.

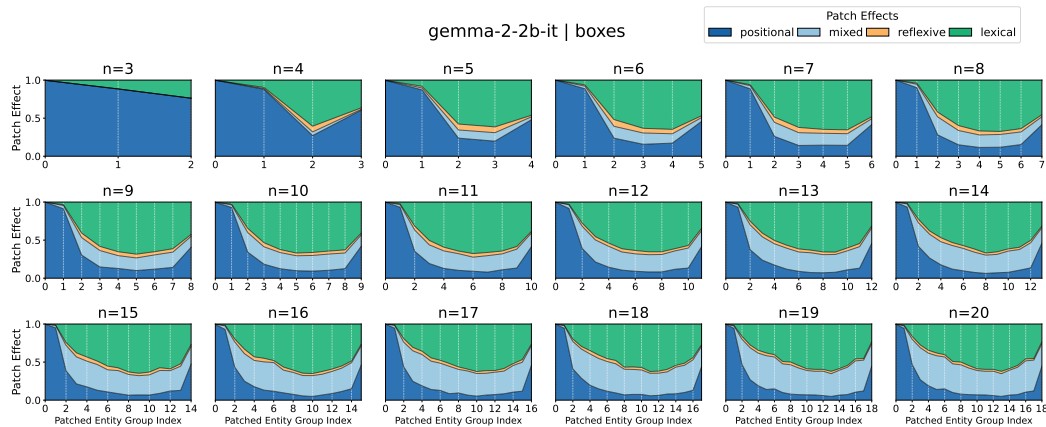

Figure 14: Results for the `TargetRebind` interchange intervention on gemma-2-2b-it for $n \in [3, 20]$ and $t_{\text{entity}} = 3$ on the *boxes* task. We see a trend where, the more entity groups need to be bound in context, the worse the positional mechanism is at binding those in the middle.

the counterfactual example is at or below random chance. However, for $\ell + 1$ and $\ell + 2$ this effect becomes the majority, showing that the model has shifted to retrieval. We also see that this layer is consistent across tasks in Figures 10 and 11.

## D.3 REMOVING TARGETED ENTITY TOKENS

In §3.1 we detail how the lexical and reflexive mechanisms are pointers that get dereferenced to the queried entity. To strengthen these claims, in this section, we evaluate what happens when we modify the `TargetRebind` interchange intervention, such that the entities targeted by those mechanisms do not exist in the original prompt. Thus, for the example in Figure 1, to test the lexical

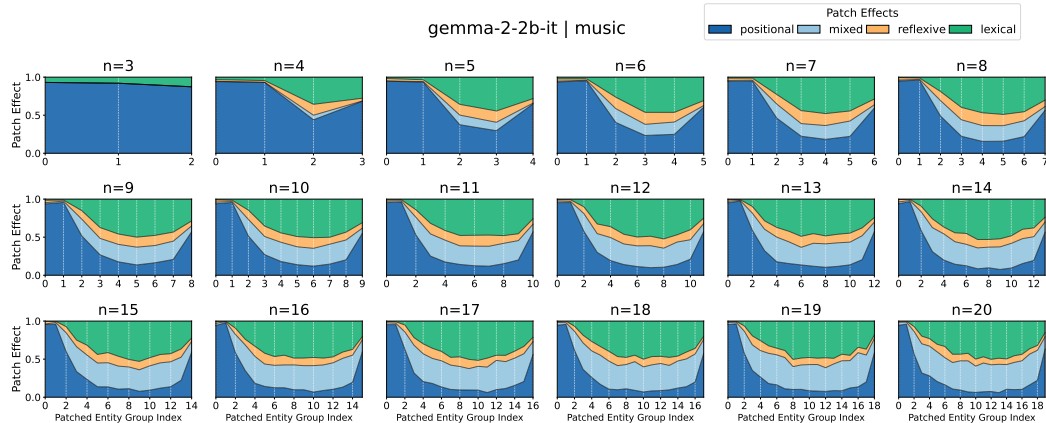

Figure 15: Results for the `TargetRebind` interchange intervention on gemma-2-2b-it for $n \in [3, 20]$ and $t_{entity} = 3$ on the *music* task. We see a trend where, the more entity groups need to be bound in context, the worse the positional mechanism is at binding those in the middle.

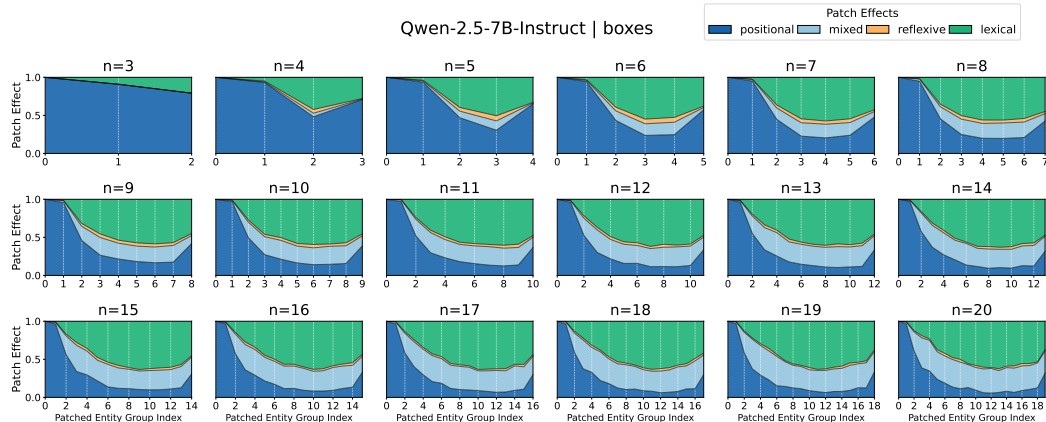

Figure 16: Results for the `TargetRebind` interchange intervention on qwen2.5-7b-it for $n \in [3, 20]$ and $t_{entity} = 3$ on the *boxes* task. We see a trend where, the more entity groups need to be bound in context, the worse the positional mechanism is at binding those in the middle.

mechanism we'd change the counterfactual such that *Ann* is replaced with a different new name *Max*, and for the reflexive we'd change *pie* to *cod* (separately). We see in Figure 28 that this leads the model to rely solely on the positional mechanism, since the others have pointers that cannot be dereferenced. In Figure 29 we see that in this case, relying on the positional mechanism yields a noisy distribution around the positional index.

A possible alternative explanation for why the model isn't retrieving the entity pointed to by these two mechanisms, is that there might be some other mechanism that prevents the model from answering with entities that do not exist in the context. To evaluate this, we conduct the same exact interventions, but for layer $\ell + 1$, where the retrieval is already taking place (see §D.2). Thus, if such a mechanism exists, we'd expect to see the same results, where the model relies solely on the positional mechanism. Otherwise, we'd expect the model to respond with the retrieved answer from the counterfactual. We can see in Figure 30 that the model indeed responds with the retrieved answer from the counterfactual, falsifying this alternative explanation. Thus, we conclude that the model indeed relies on the lexical and reflexive mechanisms as pointers for dereferencing.

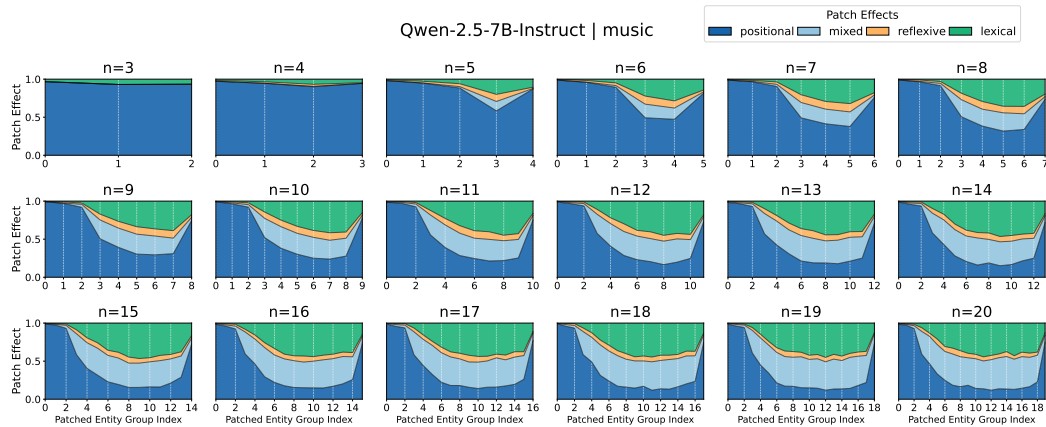

Figure 17: Results for the `TargetRebind` and $t_{\text{entity}} = 3$ interchange intervention on qwen2.5-7b-it for $n \in [3, 20]$ on the *music* task. We see a trend where, the more entity groups need to be bound in context, the worse the positional mechanism is at binding those in the middle.

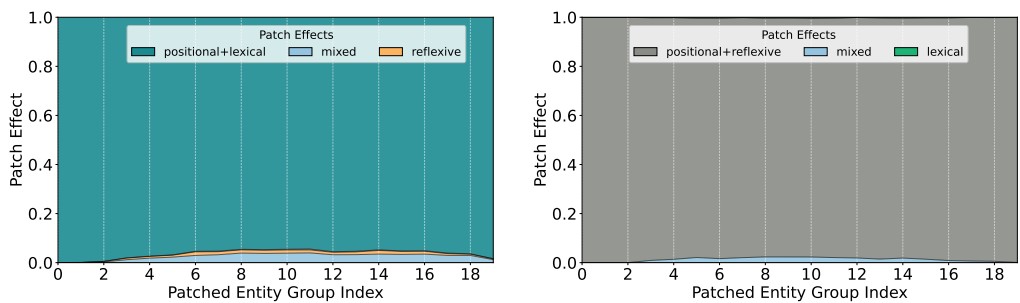

Figure 18: We evaluate gemma-2-2b-it's behavior when aligning the mechanisms for the *music* task. We align the positional and reflexive mechanisms for $t_{\text{entity}} = 1$, and the positional and lexical mechanisms for $t_{\text{entity}} = 3$. We see that when the mechanisms point at the same entity for retrieval, the model consistently responds with the correct entity.

### D.4 LINGUISTIC VARIABILITY

To assess our findings' generalization beyond the templatic datasets defined in Table 1, we incorporate linguistic variations in the phrasings of each entity group. We define 12 such variations for the *boxes* and *music* tasks respectively, such that when creating a prompt from a binding matrix $\mathbf{G}$, we choose a random variation per entity group. For example, in the *boxes* task, we have variations like "the Object is stored in box Box", "the Object was left in box Box" and "the Object ended up in box Box". We can see in the results, shown in Figure 21, that our findings remain identical in this setting. As in previous results, the model relies on the three mechanisms, mediated by the entity group index as well as $t_{\text{entity}}$.

### E   ADDITIONAL CAUSAL MODELS

We report the KL divergence scores for gemma-2-2b-it on the *music* task in Table 3. We additionally report all metrics for gemma-2-2b-it on the *sports* task in Table 4, and qwen2.5-7b-it on both tasks in Tables 5 and 6. For training, we use Adam ($\beta_1 = 0.9$, $\beta_2 = 0.999$) with learning rate 0.05, run for up to 2,000 epochs with a batch size of 512 and early stopping after 200 epochs.

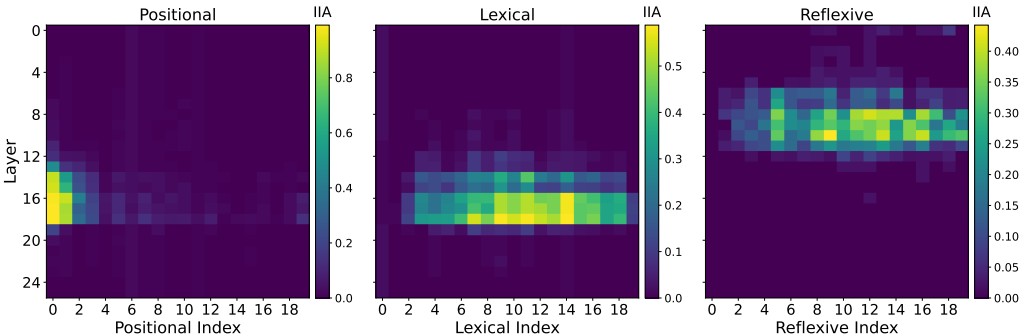

Figure 19: Results for the `PosSwap` (left), `LexSwap` (middle) and `RefSwap` (right) interchange interventions on gemma-2-2b-it for the *boxes* task. Each square shows the interchange intervention accuracy (IIA) for a given layer and positional, lexical or reflexive index. We see that positional and lexical binding information exists in entity tokens in layers 12-19, while reflexive binding information does in layers 6-12.

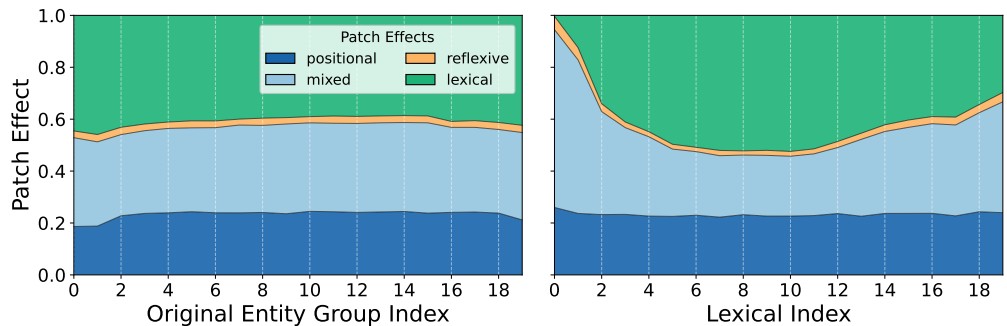

Figure 20: We show results of the `TargetRebind` interchange intervention on gemma-2-2b-it for the *boxes* task with different indices on the x-axis. **Left**: using the index of the queried entity group. This has little effect overall, except for dips at the first and last indices in the positional effect. Under `TargetRebind`, the positions of queried entity groups cannot coincide between the counterfactual and original prompts. Thus, when the original query targets the first or last groups—where positional information is strongest—these groups are never patched, slightly weakening results on average. **Right**: using the lexical index. Here the pattern mirrors Figure 3, with weaker effects at the edges and stronger ones in the middle.

.

## F FURTHER VALIDATION OF THE REFLEXIVE MECHANISM

In §3.1, we describe the reflexive binding mechanism, where a direct pointer originating from an entity token is used to point back at itself. In this section we provide further evidence for the existence of this mechanism as described.

We do this by knocking out attention from the last token position to the target entity (Geva et al., 2023), shown in Figure 23. Again we see that the model does not respond with the counterfactual target entity unless it can find it in context, which we prevent by blocking attention to it. Conversely, blocking attention at a layer when the model has already retrieved the answer, while patching at that layer, does not prevent the model from answering with the answer entity from the counterfactual. Thus, we can conclude that the model indeed relies on a reflexive mechanism for binding and retrieving entities in context.

| Name | Original | Counterfactual | Patch Positions | Patch Effects |
|------|----------|----------------|-----------------|---------------|
| `TargetRebind` | The bottle is in box *C*, the *pen* is in box A, the *ball* is in box *Q*, and the **rock** is in box N. Which box is the **rock** in? | The bottle is in box *Q*, the *ball* is in box A, the **pen** is in box *C*, and the rock is in box N. Which box is the **pen** in? | Last token position | Q: Positional A: Lexical C: Reflexive N: No effect |
| `PosSwap` | The *pen* is in box *A* and the ***ball*** is in box *Q*. Which box is the **ball** in? | The ***ball*** is in box *Q* and the *pen* is in box *A*. Which box is the **ball** in? | A→A    Q→Q | A: Patched tokens encode positional binding used by the model Q: Patched tokens do not encode positional binding used by the model |
| `LexSwap` | The *pen* is in box A and the ***ball*** is in box Q. Which box is the **ball** in? | The ***ball*** is in box A and the *pen* is in box Q. Which box is the **ball** in? | A→A    Q→Q | A: Patched tokens encode lexical binding used by the model Q: Patched tokens do not encode lexical binding used by the model |
| `RefSwap` | The *pen* is in box A and the ***ball*** is in box Q. What is in **Box Q**? | The ***ball*** is in box A and the *pen* is in box Q. What is in **Box Q**? | A→A    Q→Q | A: Patched tokens encode reflexive binding used by the model Q: Patched tokens do not encode reflexive binding used by the model |

Table 2: Original/counterfactual pair examples for interchange interventions.

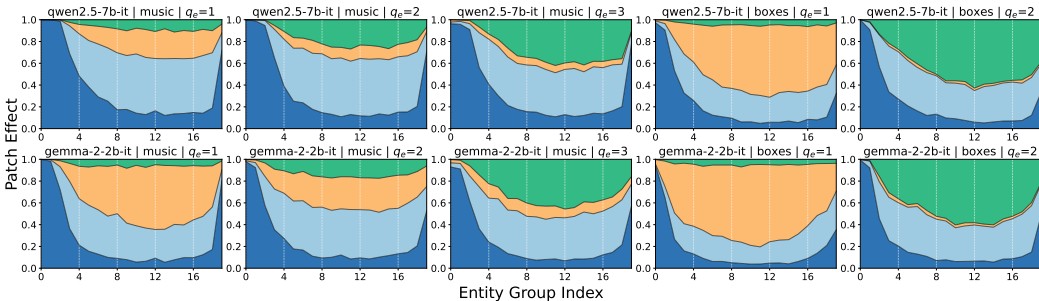

Figure 21: Results for the `TargetRebind` interchange intervention on qwen2.5-7b-it and gemma-2-2b-it for all values of $q_{\text{entity}}$ on the *boxes* and *music* binding tasks, while using random linguistic variations for the phrasings of each entity group. We find that our findings remain consistent in this setting as well.

## G    CONTEXT LENGTH ABLATION STUDY

In our evaluations, we show the effect of the number of entities that need to be bound in-context on LMs' use of the positional, lexical and reflexive mechanisms. However, a confounding factor is that as the number of entities increases, so does the length of the sequence itself. To disentangle these effects, we pad contexts with $n \in [3, 19]$ so that all sequences match the length of those with $n = 20$. Padding is done using "entity-less" sentences, as described in §5. The results are shown

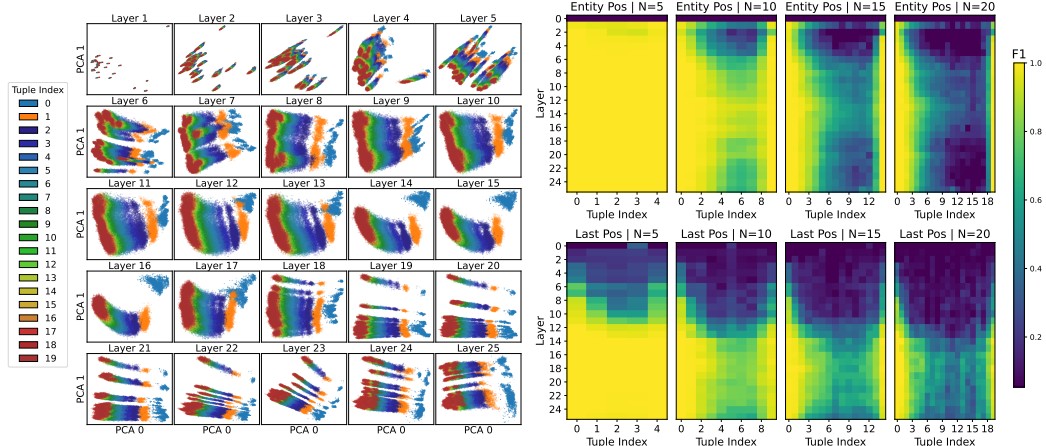

Figure 22: Separability of hidden states for entity token positions and the last token position, across layers and values of $n$. PCA projections (left) and multinomial logistic regression probes (right) show that first and last entity groups are linearly separable, while middle groups overlap substantially. Separability decreases as the number of entities $n$ increases.

| Model | $\mathbf{KL}_{t\|p} \downarrow$ | | | $\mathbf{KL}_{p\|t} \downarrow$ | | |
|---|---|---|---|---|---|---|
| | $t_{\text{entity}} = 1$ | $t_{\text{entity}} = 2$ | $t_{\text{entity}} = 3$ | $t_{\text{entity}} = 1$ | $t_{\text{entity}} = 2$ | $t_{\text{entity}} = 3$ |
| $\mathcal{M}$ | **0.22** | **0.17** | **0.26** | 0.31 | **0.21** | **0.41** |
| $\mathcal{M}$ w/ Oracle Pos | 0.14 | 0.08 | 0.14 | 0.32 | 0.11 | 0.24 |
| $\mathcal{M}$ w/ One-Hot Pos | 0.71 | 0.67 | 0.71 | 1.00 | 0.88 | 0.88 |
| Only One-Hot At Pos | 6.41 | 5.61 | 5.95 | 3.41 | 2.39 | 2.78 |
| $\mathcal{M} \setminus \{P\}$ | 1.75 | 1.52 | 1.71 | 4.51 | 2.37 | 3.17 |
| $\mathcal{M} \setminus \{L\}$ | 0.39 | 0.73 | 1.76 | **0.3** | 0.37 | 1.42 |
| $\mathcal{M} \setminus \{R\}$ | 2.14 | 1.08 | 0.61 | 2.10 | 0.54 | 0.44 |
| $\mathcal{M} \setminus \{L, R\}$ | 2.10 | 1.22 | 1.82 | 2.13 | 0.73 | 1.50 |
| $\mathcal{M} \setminus \{P, R\}$ | 9.19 | 7.35 | 5.32 | 10.7 | 5.55 | 4.34 |
| $\mathcal{M} \setminus \{P, L\}$ | 4.66 | 6.18 | 8.45 | 4.28 | 2.92 | 5.40 |
| Uniform | 2.71 | 1.96 | 2.44 | 7.57 | 3.49 | 4.84 |

Table 3: KL divergence results for modeling an LM's behavior contingent on the positional, lexical and reflexive indices. Evaluated on gemma-2-2b-it for the *music* binding task. Our full model achieves the best performance, only slightly below the oracle.

in Figure 27. If the effects of increasing $n$ were due to increasing sequence length, we'd expect all results to be identical to when setting $n = 20$, and to each other. However, we see that, while the distribution of patch effects is slightly affected by padding, the results and trends align closely with our results without padding. This indicates that model behavior is governed primarily by the number of entities that must be bound, rather than by sequence length.

# H  LLM USAGE

In this work, the authors relied on LLMs solely to assist with implementing specific helper functions.

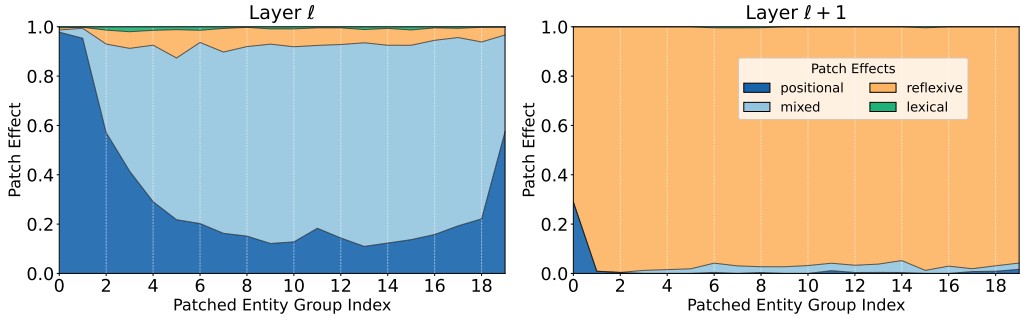

Figure 23: Patch effects under `TargetRebind` for gemma-2-2b-it while blocking attention to the target entity. Left: blocking attention when the model is accumulating binding information in the last token position leads to it not being able to dereference the reflexive pointer. Had the patch contained the retrieved answer, this plot would be fully orange. Right: patching at the following layer and blocking attention to the target entity. Here the plot is fully orange since the entity has already been retrieved.

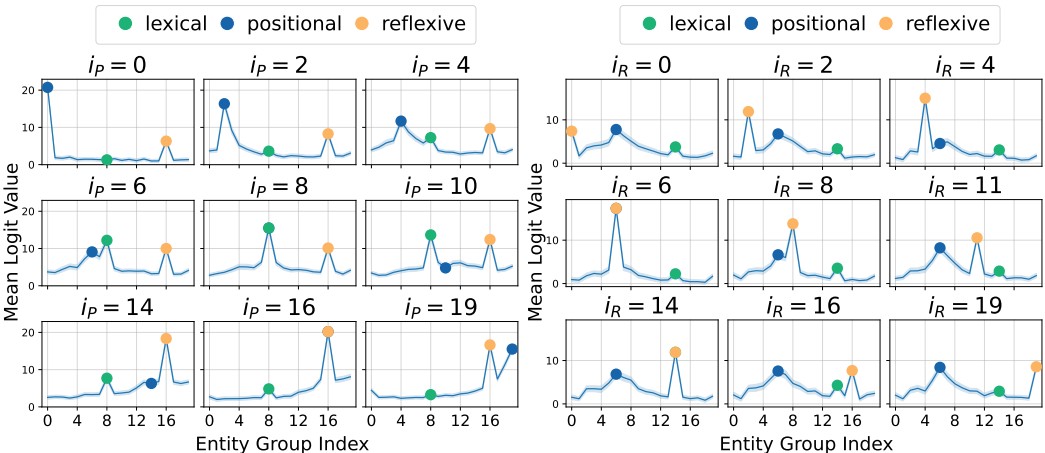

Figure 24: Mean logit distributions under `TargetRebind` for gemma-2-2b-it on the *music* task. Left: fixing $i_L = 8, i_R = 16$ and varying $i_P$. Right: fixing $i_P = 6, i_L = 14$ and varying $i_R$.

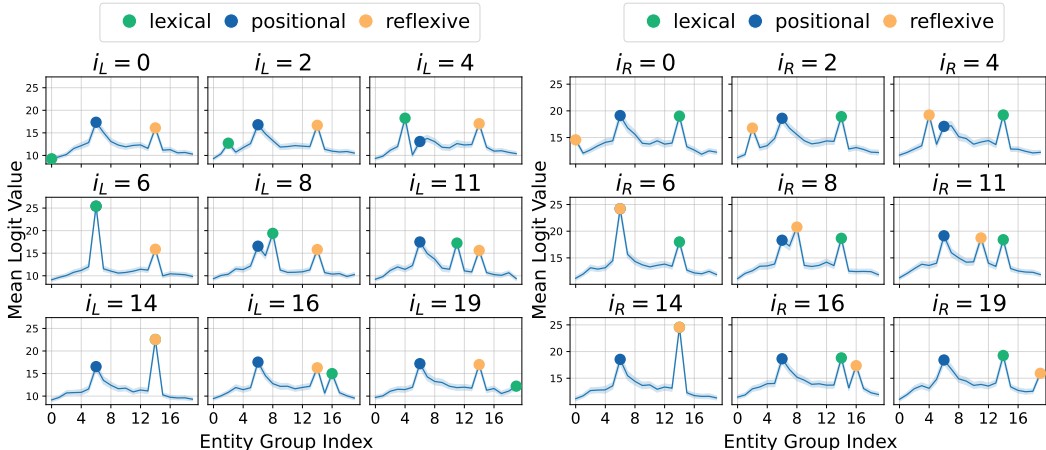

Figure 25: Mean logit distributions under `TargetRebind` for qwen2.5-7b-it on the *music task*. Left: fixing $i_P = 6, i_R = 14$ and varying $i_L$. Right: fixing $i_P = 6, i_L = 14$ and varying $i_R$.

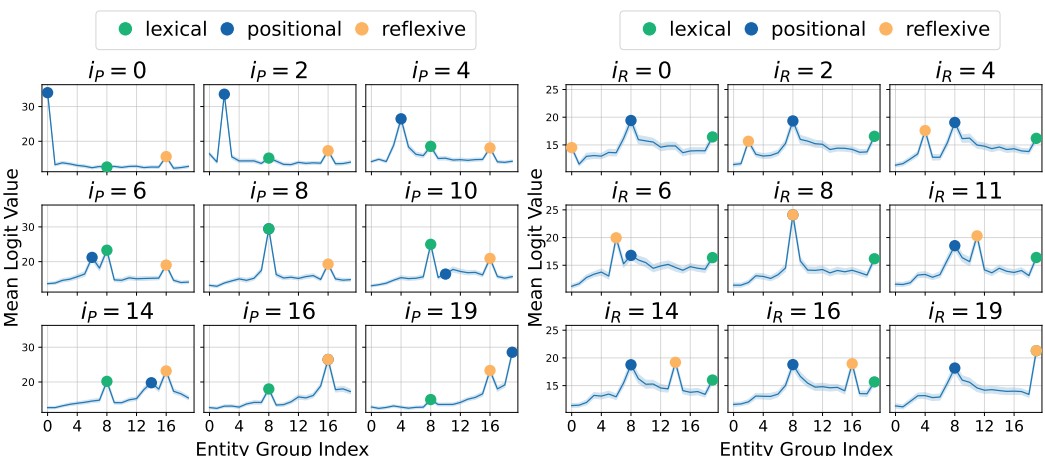

Figure 26: Mean logit distributions under TargetRebind for qwen2.5-7b-it on the *sports* task. Left: fixing $i_L = 8, i_R = 16$ and varying $i_P$. Right: fixing $i_P = 8, i_L = 19$ and varying $i_R$.

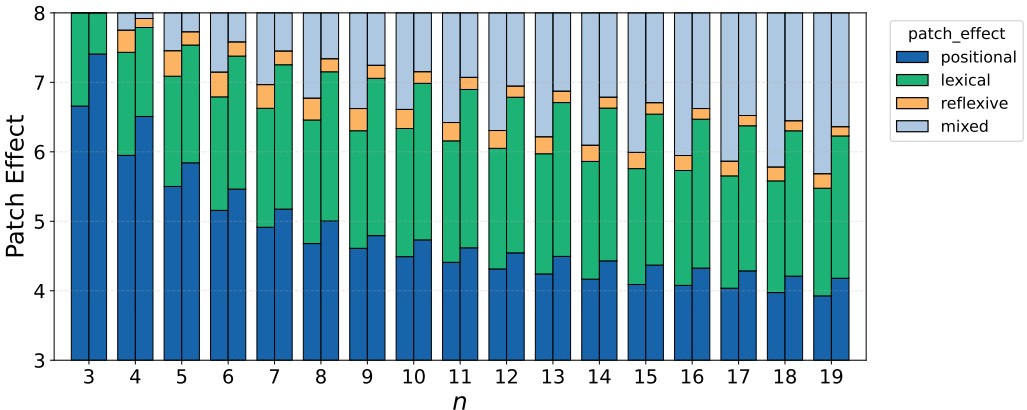

Figure 27: Mean patch effects per number of entities in context ($n$). For each $n$, we report the standard mean patch effects (right) alongside results from padded sequences (left), where sequence length is fixed to match $n = 20$. While padding slightly shifts the distribution of patch effects, the overall patterns remain consistent: model behavior is primarily controlled by the number of entities in context, rather than sequence length.

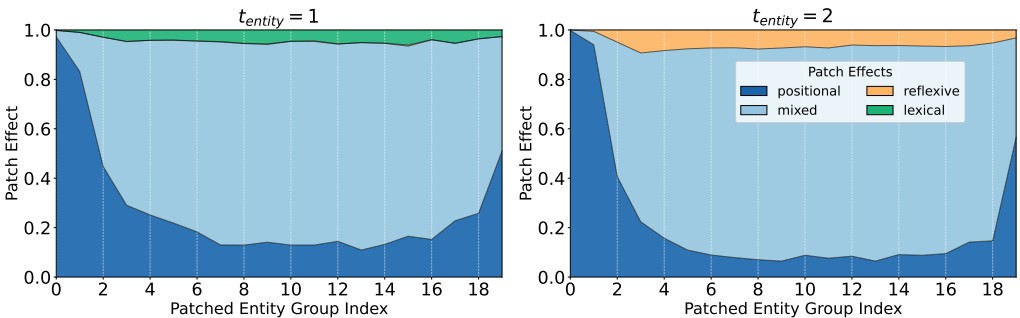

Figure 28: Left: results for `TargetRebind` interchange intervention on gemma-2-2b-it with $t_{\text{entity}} = 1$, where the query entity in the counterfactual does not exist in the original. Right: results for `TargetRebind` interchange intervention on gemma-2-2b-it with $t_{\text{entity}} = 2$, where the target entity in the counterfactual does not exist in the original. We can see in both plots that when the model can't use the lexical and reflexive mechanisms since the entities they point to don't exist, the model falls back to solely using the positional mechanism (distribution showed in Figure 29).

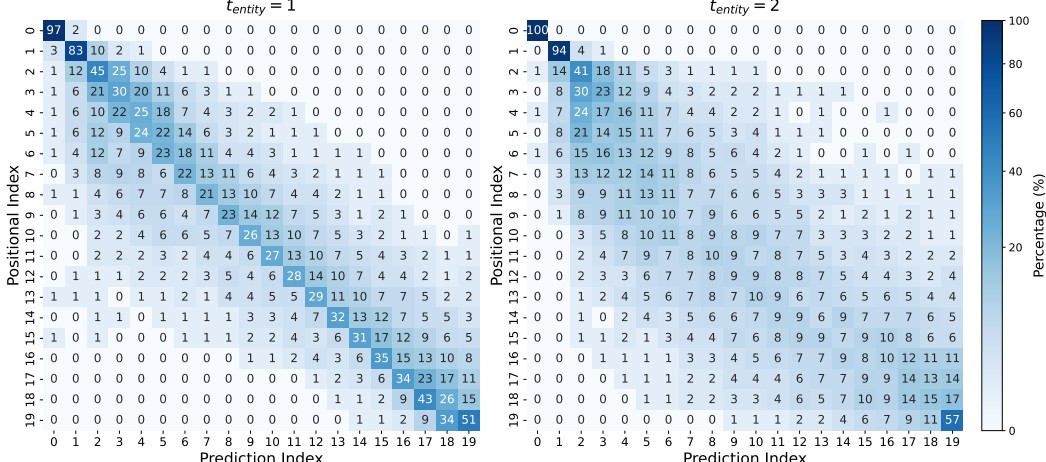

Figure 29: Left: confusion matrix for non-lexical and reflexive patch effects under the `TargetRebind` interchange intervention on gemma-2-2b-it with $t_{\text{entity}} = 1$, where the query entity in the counterfactual does not exist in the original. Right: results for non-lexical and reflexive patch effects under the `TargetRebind` interchange intervention on gemma-2-2b-it with $t_{\text{entity}} = 2$, where the queried entity in the counterfactual does not exist in the original. We can see that when the model can't use the lexical and reflexive mechanisms, it falls back on the noisy positional mechanism.

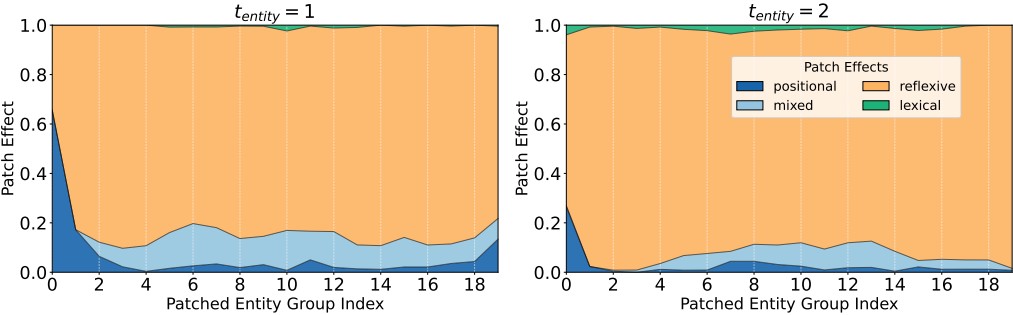

Figure 30: Results for `TargetRebind` interchange interventions on gemma-2-2b-it with $t_{\text{entity}} \in [2]$, where the query entity (left) or queried entity (right) in the counterfactual do not exist in the original, patching at layer $\ell + 1$. We see that the model copies the retrieved answer from the counterfactual, showing that no mechanism exists to suppress answering with entities that do not exist in the original prompt.

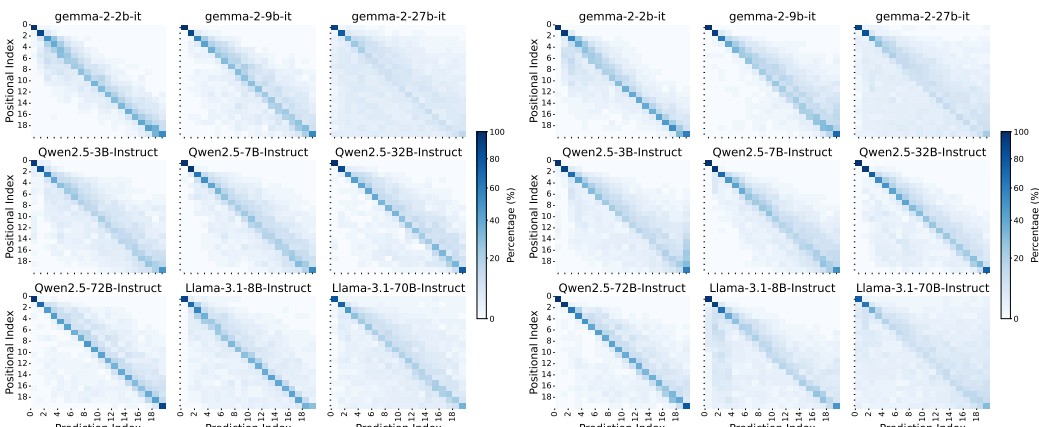

Figure 31: Confusion matrix for non-lexical and reflexive patch effects under the `TargetRebind` interchange intervention for all models, showing the diffuse distribution around the positional index. Left: $t_{\text{entity}} = 1$. Right: $t_{\text{entity}} = 2$.

| Model | JSS $\uparrow$ | | | $\mathbf{KL}_{t\|p} \downarrow$ | | | $\mathbf{KL}_{p\|t} \downarrow$ | | |
|---|---|---|---|---|---|---|---|---|---|
| | $t=1$ | $t=2$ | $t=3$ | $t=1$ | $t=2$ | $t=3$ | $t=1$ | $t=2$ | $t=3$ |
| $\mathcal{M}$ ($L_{\text{one-hot}}$, $R_{\text{one-hot}}$,$P_{\text{Gauss}}$) | **0.94** | **0.95** | **0.93** | **0.3** | **0.21** | **0.31** | **0.35** | **0.28** | **0.39** |
| $\mathcal{M}$ w/ $P_{\text{oracle}}$ | 0.96 | 0.97 | 0.96 | 0.13 | 0.11 | 0.13 | 0.32 | 0.19 | 0.21 |
| $\mathcal{M}$ w/ $P_{\text{one-hot}}$ | 0.85 | 0.87 | 0.87 | 0.77 | 0.59 | 0.62 | 1.2 | 0.81 | 0.73 |
| $\mathcal{P}_{\text{one-hot}}$ (prevailing view) | 0.4 | 0.46 | 0.43 | 6.5 | 5.74 | 6.03 | 3.72 | 2.54 | 2.9 |
| $\mathcal{M} \setminus \{P\}$ | 0.67 | 0.69 | 0.71 | 1.75 | 1.53 | 1.49 | 4.91 | 3.04 | 1.88 |
| $\mathcal{M} \setminus \{L\}$ | 0.93 | 0.89 | 0.75 | 0.41 | 0.8 | 1.99 | 0.37 | 0.53 | 1.25 |
| $\mathcal{M} \setminus \{R\}$ | 0.69 | 0.84 | 0.9 | 1.83 | 1.28 | 1.04 | 2.52 | 0.71 | 0.47 |
| $\mathcal{M} \setminus \{L, R\}$ | 0.68 | 0.79 | 0.74 | 1.84 | 1.52 | 2.06 | 2.54 | 1.05 | 1.36 |
| $\mathcal{M} \setminus \{P, R\}$ | 0.11 | 0.31 | 0.47 | 9.25 | 7.19 | 5.32 | 10.9 | 6.11 | 3.61 |
| $\mathcal{M} \setminus \{P, L\}$ | 0.55 | 0.45 | 0.23 | 4.71 | 5.95 | 8.16 | 4.64 | 3.06 | 4.49 |
| Uniform | 0.45 | 0.54 | 0.54 | 2.66 | 2.13 | 2.22 | 8 | 4.78 | 2.93 |

Table 4: Results for modeling gemma-2-2b-it's behavior on the *sports* binding task, contingent on the positional, lexical and reflexive indices. Here $t$ denotes $t_{\text{entity}}$.

| Model | JSS $\uparrow$ | | | $\mathbf{KL}_{t\|p} \downarrow$ | | | $\mathbf{KL}_{p\|t} \downarrow$ | | |
|---|---|---|---|---|---|---|---|---|---|
| | $t=1$ | $t=2$ | $t=3$ | $t=1$ | $t=2$ | $t=3$ | $t=1$ | $t=2$ | $t=3$ |
| $\mathcal{M}$ ($L_{\text{one-hot}}$, $R_{\text{one-hot}}$,$P_{\text{Gauss}}$) | **0.94** | **0.92** | **0.92** | **0.27** | **0.34** | **0.37** | 0.35 | **0.48** | **0.45** |
| $\mathcal{M}$ w/ $P_{\text{oracle}}$ | 0.98 | 0.98 | 0.98 | 0.07 | 0.07 | 0.07 | 0.1 | 0.1 | 0.1 |
| $\mathcal{M}$ w/ $P_{\text{one-hot}}$ | 0.87 | 0.89 | 0.88 | 0.58 | 0.47 | 0.53 | 1.09 | 0.74 | 0.69 |
| $\mathcal{P}_{\text{one-hot}}$ (prevailing view) | 0.56 | 0.55 | 0.53 | 4.66 | 4.65 | 5.01 | 1.84 | 1.88 | 2.04 |
| $\mathcal{M} \setminus \{P\}$ | 0.62 | 0.66 | 0.66 | 1.85 | 1.68 | 1.73 | 5.05 | 3.29 | 2.55 |
| $\mathcal{M} \setminus \{L\}$ | 0.93 | 0.85 | 0.78 | 0.51 | 1.14 | 1.77 | **0.33** | 0.74 | 1.09 |
| $\mathcal{M} \setminus \{R\}$ | 0.86 | 0.9 | 0.9 | 0.9 | 0.89 | 1.0 | 0.68 | 0.54 | 0.5 |
| $\mathcal{M} \setminus \{L, R\}$ | 0.86 | 0.84 | 0.78 | 0.92 | 1.24 | 1.79 | 0.71 | 0.83 | 1.11 |
| $\mathcal{M} \setminus \{P, R\}$ | 0.16 | 0.34 | 0.42 | 8.43 | 6.69 | 5.93 | 9.04 | 5.6 | 4.4 |
| $\mathcal{M} \setminus \{P, L\}$ | 0.35 | 0.24 | 0.17 | 6.64 | 7.8 | 8.57 | 6.8 | 5.3 | 5.32 |
| Uniform | 0.55 | 0.58 | 0.54 | 2.11 | 1.95 | 2.21 | 5.92 | 4.09 | 3.54 |

Table 5: Results for modeling qwen2.5-7b-it's behavior on the *music* binding task, contingent on the positional, lexical and reflexive indices. Here $t$ denotes $t_{\text{entity}}$.

| Model | JSS ↑ | | | $\mathbf{KL}_{t\|p}$ ↓ | | | $\mathbf{KL}_{p\|t}$ ↓ | | |
|---|---|---|---|---|---|---|---|---|---|
| | $t=1$ | $t=2$ | $t=3$ | $t=1$ | $t=2$ | $t=3$ | $t=1$ | $t=2$ | $t=3$ |
| $\mathcal{M}$ ($L_{\text{one-hot}}$, $R_{\text{one-hot}}$,$P_{\text{Gauss}}$) | **0.95** | **0.93** | **0.92** | **0.24** | **0.31** | **0.36** | 0.28 | **0.39** | **0.47** |
| $\mathcal{M}$ w/ $P_{\text{oracle}}$ | 0.98 | 0.98 | 0.97 | 0.07 | 0.08 | 0.11 | 0.09 | 0.11 | 0.18 |
| $\mathcal{M}$ w/ $P_{\text{one-hot}}$ | 0.87 | 0.89 | 0.88 | 0.62 | 0.52 | 0.55 | 1.14 | 0.68 | 0.84 |
| $\mathcal{P}_{\text{one-hot}}$ (prevailing view) | 0.57 | 0.55 | 0.51 | 4.58 | 4.72 | 5.18 | 1.73 | 1.84 | 2.21 |
| $\mathcal{M} \setminus \{P\}$ | 0.61 | 0.66 | 0.66 | 1.88 | 1.75 | 1.73 | 5.11 | 2.9 | 3.35 |
| $\mathcal{M} \setminus \{L\}$ | 0.94 | 0.87 | 0.77 | 0.53 | 1.09 | 1.64 | **0.27** | 0.64 | 1.27 |
| $\mathcal{M} \setminus \{R\}$ | 0.87 | 0.89 | 0.91 | 0.89 | 1.05 | 0.85 | 0.57 | 0.54 | 0.48 |
| $\mathcal{M} \setminus \{L, R\}$ | 0.87 | 0.83 | 0.77 | 0.92 | 1.33 | 1.65 | 0.6 | 0.82 | 1.29 |
| $\mathcal{M} \setminus \{P, R\}$ | 0.17 | 0.31 | 0.44 | 8.39 | 7.01 | 5.75 | 9.04 | 5.54 | 4.78 |
| $\mathcal{M} \setminus \{P, L\}$ | 0.33 | 0.32 | 0.14 | 6.77 | 7.27 | 8.77 | 6.97 | 3.88 | 7.27 |
| Uniform | 0.54 | 0.56 | 0.53 | 2.12 | 2.06 | 2.25 | 6.01 | 3.9 | 4.71 |

Table 6: Results for modeling qwen2.5-7b-it's behavior on the *sports* binding task, contingent on the positional, lexical and reflexive indices. Here $t$ denotes $t_{\text{entity}}$.

