# OpenReview forum: "Mixing Mechanisms: How Language Models Retrieve Bound Entities In-Context"
_ICLR.cc/2026/Conference — ICLR 2026 Poster_

### Official Review · Reviewer_4tWV · 2025-10-24

**Soundness:** 4
**Presentation:** 4
**Contribution:** 3
**Rating:** 8
**Confidence:** 3

**Summary:**

This paper investigates how LLMs bind and retrieve entities in context. Prior work has suggested that LLMs rely mainly on a positional mechanism, retrieving entities by their relative position in a list of contextually bound entities. However, the authors show that this mechanism becomes unreliable when the context grows and entities appear in middle positions, reminiscent of the lost-in-the-middle effect. To explain this, the paper proposes lexical and reflexive mechanisms that complement the positional one. Through systematic interchange interventions, they find that LLMs dynamically mix these three mechanisms depending on position and entity type. They further build a causal model that combines all three mechanisms and achieves up to 95% Jensen–Shannon similarity with true model predictions.

**Strengths:**

* The authors propose the lexical and reflexive mechanisms as natural extensions of how binding might operate when positional cues become unreliable. They support these hypotheses with well-designed counterfactual interventions that yield clear empirical evidence. The experimental setup, including controlled manipulations of entity positions and roles, provides high causal interpretability rather than correlational evidence.
* The findings are replicated across nine model families and ten distinct tasks,  demonstrating robustness. The mixture model achieves 95% agreement with LLM token predictions, further demonstrating the faithfulness of the proposed framework.
* The parallels drawn to primacy and recency biases in human memory, and the connection to the lost-in-the-middle effect, make the results conceptually relatable.

**Weaknesses:**

* The three mechanisms are inferred largely through counterfactual patching, but the causal independence between them is assumed rather than rigorously established. For example, lexical and reflexive mechanisms often co-occur. It remains unclear if they are distinct causal variables or correlated manifestations of shared attention dynamics.
* Although filler text is introduced later, most analyses still rely on templated X likes Y style prompts. These may not capture linguistic variability or discourse-level entity binding.

**Questions:**

See Weaknesses.

---

> ### Author Response · Authors · 2025-11-21
>
> We thank the reviewer for their favorable review, and are pleased they found our methodology well-designed, and our results robust.
>
> **W1: Causal Independence**
> We are not sure exactly what the reviewer meant by ‘causal independence’ or ‘correlated manifestations’.
>
> If the reviewer meant that we don’t show that each mechanism can affect the model’s output independently, then we contend that our experiments do in fact show this. In our counterfactual design, counterfactuals are crafted such that an interchange intervention will lead to each mechanism pointing to a different entity. This allows us to causally verify the existence and independence of each of the mechanisms. Figures 2 and 4 show that each mechanism affects the model differently by promoting different entities, and that these mechanisms are active in different conditions (e.g. target entity). We also conduct experiments where we fix two mechanisms to point at two fixed entities, while causally affecting and shifting only the third mechanism (Figures 3, 13, 24-26). These show that each mechanism is distinct and can be causally controlled independently of one another.
>
> If, however, the reviewer alluded to evaluating whether the internal components used to store and manipulate these mechanisms and their signals (e.g. attention heads, residual stream subspaces) are independent, then we agree with their point. These evaluations would be very interesting to conduct, and could hold promising insights regarding how models store signals, and how model components read, write and manipulate them. But, since they would involve many more evaluations and perhaps new experimental methodologies (such as feature localization and circuit discovery), and as our work is already quite dense with evaluations, we leave this to future work. We do however stress that the causal independence of the three mechanisms, which we show conclusively in our work, is not contingent on the results of such evaluations. For example, a single attention head’s attention scores could be affected by all three mechanisms at once, while each mechanism can still causally affect it differently and independently.
>
> **W2: Linguistic Variability**
> We agree with the reviewer’s point regarding the templatic nature of our datasets. As the reviewer notes, we address this partially in Section 5 by mimicking discourse-type text through interleaving entity groups with an increasing amount of text, showing that our findings stay consistent.
>
> To address the matter of linguistic variability, we add a new evaluation in Appendix Section D.4 wherein we vary the phrasings of each entity group randomly. We do this by defining 12 variations for textualizing entity groups per binding task. Thus, each entity group in a given prompt is phrased randomly according to one of the 12 variations. For example, in the *boxes* binding task we have `”the {Object} is stored in box {Box}”`, `”the {Object} was left in box {Box}”`, `”the {Object} ended up in box {Box}”` etc. These variations enable us to conduct our experiments on a less templatic and more linguistically variable dataset. We conduct this evaluation for gemma-2-2b-it and qwen2.5-7b-it for the *boxes* and *music* tasks, and show that our findings remain remarkably consistent.
>
> In addition to this evaluation, we think it would be valuable to conduct evaluations that extend these counterfactuals even further. For example, an evaluation where the entities’ orderings within a group are shuffled per entity group could enable us to further establish the robustness of our findings. However, these evaluations would require designing wholly new counterfactual designs and interchange interventions which would allow us to study this while separating and controlling for the effect of each entity’s order of appearance in the original and counterfactual prompts. We therefore think that this is best left for future work.
>
> ---
>
> We hope that we have addressed the reviewer’s points sufficiently, and thank them for their time. We are happy to discuss these issues further, or incorporate any additional comments the reviewer may have.

---

> > ### Author Response · Authors · 2025-11-26
> >
> > Dear reviewer 4tWV,
> >
> > Thank you again for your positive review and insightful suggestions. We have posted a response addressing your points in detail. We discussed the causal independence of the three mechanisms, and added an experiment incorporating linguistic variability as per your suggestion to the revised version of our paper.
> >
> > Thank you again for your time, and we would be happy to address any further questions you may have.

---

### Official Review · Reviewer_U696 · 2025-10-30

**Soundness:** 3
**Presentation:** 3
**Contribution:** 4
**Rating:** 8
**Confidence:** 4

**Summary:**

The authors explore multiple mechanisms by which transformers LMs track entities in context to retrieve factual information. It's already known that models will track the positions of entities and use this mechanism to answer factual questions. The positional mechanism, however, breaks down when there are many entities in context. The authors find that transformers have two other mechanisms: a lexical mechanism where the model looks up the queried entity and a reflexive mechanism where a model looks up information about a promoted token. These mechanisms can all disagree about the answer to a query, which the authors show with interventional experiments. They also create an effective model of this three-mechanism behavior that closely matches the output distributions of the LMs they test on.

**Strengths:**

- It seems intuitive that something like the lexical mechanism or the reflexive mechanism would exist. It's somewhat surprising that these are separate.
- The interventions show that the different retrieval mechanisms can disagree in their predictions and behavior on the output distribution, which is compelling.
- The authors create an effective model of the three retrieval mechanisms which closely matches the output distribution of the actual model.
- Polluting the context with additional free-form text is a reasonable robustness check

**Weaknesses:**

- I'm looking for more details about how exactly the interchange interventions were performed and how the authors localized where to do interchange interventions. From the appendix, it seems like this is based on performing interchange interventions on the attention at specific layers on the last token position? And the authors use attention knockout to figure out which layers are passing entity information to the final token position? Can you clarify?
- The main body of the paper would benefit from a bit more explanation of what exactly is going on here (I'm aware of space constraints, but I think this is important). And the appendix would benefit from a clear, high-level description of what the experiments are actually doing.

Minor suggestions and feedback:
- Putting the counterfactual input on top in Figure 1 feels confusing, seems more natural to have the original on top and the counterfactual below?
- I found the text description of the reflexive mechanism (line 231) to be fairly hard to parse. This seems like something that is much easier to show than to say, so a figure could be helpful.
- Line 159: use \citet instead of \citep

**Questions:**

- It seems like the model uses the reflexive pathway as a "verification" step. Does the reflexive pathway need to match the entity from the lexical pathway (as it does in Figure 1) to work? Is my understanding of this correct?
- See "weaknesses" above re: questions about how the interchange interventions were performed

---

> ### Author Response · Authors · 2025-11-21
>
> We appreciate the reviewer’s positive review, and are happy that they found our results intuitive and compelling.
>
> **W1: Interchange Intervention Details**
> We explain the fine details of how we conduct the interchange interventions in Sections 2 and 3\. There, we explain that unless stated otherwise, all interchange interventions are conducted by patching the last token residual stream from the counterfactual prompt to the original one, at a layer $\\ell$. We focus on the last token residual stream following previous work, and validate this experimentally, where in all of our results we find that this vector indeed contains the binding signals. To localize the layer $\\ell$, identified per model, we conduct interchange interventions for each layer of the model, and identify the last layer before the model starts the retrieval of the answer entity, explained in detail in Section D.2 and shown in Figure 2 on the left.
>
> Attention knockout was not used for localization of where to conduct the interchange interventions, but rather as a tool for identifying how binding signals propagate throughout the model. These evaluations, which we include for completeness in appendix Section C, are ancillary to our main results. They are meant to build a complete picture of the model’s internals, as opposed to informing our interventions.
>
> We appreciate the reviewer’s concern regarding clarity, and so we add clarifications in Section 3 which clearly explain how we conduct our interventions. If the reviewer feels that this should be further clarified we are happy to do so.
>
> **W2: More Explanation**
> We thank the reviewer for raising this point. To improve clarity, we added additional explanations to Sections 3 and 4, specifying what each section covers and the motivation behind it. We also reorganized the appendix to better indicate the purpose of each supplementary experiment, and added introductory explanations at the start of each section describing what is done in the experiments and why. If there are specific areas the reviewer would like us to clarify further, we would be happy to do so.
>
> **W3: Suggestions and Feedback**
>
> * We agree with the reviewer that having the original input above the counterfactual one in Figure 1 would be more intuitive, and favored this option in our original design. However, that would lead the patching direction to point upwards rather than downwards, which would confuse the reader as to which direction is correct for reading the figure, since the explanatory flow is top to bottom, but the patching flow is bottom to top. Since inverting the order completely also fails to solve this issue, we felt the current organization is best, since it first shows how the mechanisms function in the model, what patching induces in it, and the resulting logit distributions. In our revised version, we also expand our caption for Figure 1 to further clarify it.
> * We thank the reviewer for this point and add a figure illustrating how the reflexive mechanism works to the revision (appendix Figure 7), which we point to from Section 3 when presenting the reflexive mechanism.
> * Fixed citation command in the revised version.
>
> **Q1: Reflexive Mechanism**
> In our paper we show that the reflexive mechanism is a separate mechanism that the model relies on for binding and retrieving entities. This finding is conveyed in the different experiments we conduct (e.g. Figures 2,3,4), but shown perhaps most clearly in Figure 3 on the right, where we show that we can set each mechanism to point at a different entity, and see that this manifests in separate peaks in the mean logit distribution over the possible answer tokens. Of course, all three mechanisms can be thought of as verifications of each other, in the sense that they’re redundant mechanisms which enable the model to effectively retrieve the correct entity.
>
> To clarify this point, we add a section in the revision (3.4) dedicated entirely to validating the existence of the reflexive mechanism, which was moved from the appendix to the main text. We hope that this clarification, along with the added figure, clarify this point sufficiently, and are happy to further elaborate on this.
>
> **Q2: Interchange Interventions**
> We address this point in W1.
>
> We thank the reviewer for their time, and hope that we addressed their concerns sufficiently. We would be happy to incorporate or discuss any additional points.

---

> > ### Author Response · Authors · 2025-11-26
> >
> > Dear reviewer U696,
> >
> > Thank you again for your positive review and constructive comments. We have posted a detailed response addressing your questions and comments, and explaining the details of our interchange interventions and the reflexive mechanism. We have also uploaded a revised version of our paper incorporating your comments: adding details to Figure 1 and Sections 3 and 4, adding a new figure explaining the reflexive mechanism (Figure 7), and adding a section diving deeper into the reflexive mechanism.
> >
> > Thank you again for your time, and if you have any remaining questions, we would be happy to address them.

---

> > > ### Comment · Reviewer_U696 · 2025-11-27
> > >
> > > **Interchange Interventions**: I think the additional explanatory details help here. The main thing I was confused about was actually whether you perform the interchange interventions on the *entire* residual stream, which does seem to be the case. I'd recommend explicitly stating this.
> > >
> > > **Reflexive Mechanism**: The additional figure is quite helpful, and I think this helps clarify my original question. There's some subtlety to how the reflexive mechanism behaves in queries of the form "Who loves pie?" (Figure 7) vs "What does Ann love?" (Figure 1). In "Who loves pie?" the reflexive mechanism is able to operate directly on "pie," but, in "What does Ann love?" it is, in some sense, still associated with Ann even though it can behave independently of the lexical mechanism under certain interchange interventions. It seems other reviewers may have had similar confusions, so some discussion of this point might be worth including. I think the addition of Section 3.4 in the main body is helpful.
> > >
> > > **Patching Figure Order**: Sure, I trust your judgment.

---

> > > > ### Author Response · Authors · 2025-12-03
> > > >
> > > > We thank the reviewer for engaging with our response and clarifications. We’re happy that our added clarifications have improved the clarity of our paper.
> > > >
> > > > To address the remaining questions, we added a clarification to Section 3 stating explicitly that we perform full vector patching, and to Section 4 discussing the LM’s reliance on the lexical and reflexive mechanisms.
> > > >
> > > > We hope that this addresses the reviewer's questions sufficiently.

---

### Official Review · Reviewer_zu1T · 2025-10-31

**Soundness:** 3
**Presentation:** 3
**Contribution:** 4
**Rating:** 8
**Confidence:** 3

**Summary:**

This paper investigates how language models (LMs) retrieve bound entities in context, e.g., “Pete loves jam, Ann loves pie. Who loves pie?”. The authors identify that retrieval arises from a mixture of three mechanisms: positional, lexical, and reflexive.
- Positional: retrieves the target based on the position of the entity group corresponding to the query.
- Lexical: retrieves the entity bound to the queried token itself (e.g., “pie” → “Ann”).
- Reflexive: uses a direct pointer to the target entity token.
They construct a controlled dataset with paired _original_ and _counterfactual_ examples (with entities shuffled), to identify each mechanism via patching experiments. This setup allows them to separate the contribution of each mechanism to the LM’s predictions. They further show that mixing these three mechanisms explains model behavior in longer contexts where the positional mechanism alone fails.
Finally, they build a simple causal model incorporating these mechanisms, which closely reproduces the LM’s next-token distribution (≈0.95 Jensen–Shannon similarity).

**Strengths:**

- **Clarity:** The paper is clearly written and well-organized. The motivation, hypotheses, and experimental setup are clearly presented with concrete examples.
- **Originality:** The discovery of two previously undescribed retrieval mechanisms (lexical and reflexive) extends the mechanistic interpretability literature beyond the known positional mechanism.
- **Quality:** The experimental methodology, which uses counterfactual patching and causal modeling, is rigorous and carefully justified.

**Weaknesses:**

- **Figures:** Some key figures (e.g., Fig. 2) are difficult to interpret. The axes, metrics, and what constitutes “mixed” effects are not clearly explained in the captions or main text, making it hard for readers to connect the visualization to the described mechanisms.
- **Presentation of reflexive mechanism:** The explanation and evidence for the reflexive mechanism remain somewhat unclear. Because the patched context may already contain the predicted token, it is hard to disentangle whether the observed effect truly demonstrates a “pointer” or simply reflects ongoing retrieval from other mechanisms.

**Questions:**

Dataset
- Could you specify the range of sizes for the entity set $\mathcal{E}$ in the dataset ?

Figures
- In Figure 2, what exactly does the _y_-axis represent? Is it the proportion of examples where that mechanism predicts the correct answer? The term “index” on the axis is confusing.
- What does “mixed” mean in this context?
- Are cases excluded where the patching does not predict a valid entity (i.e., none of the entities in the list)? Or does this never happen?

Reflexive mechanism
- In the main text, you mention patching into a context where the target entity is not present, to test the reflexive mechanism. However, it still seems possible that the patched representation already contains the target token’s activation, and that in the new context other mechanisms simply suppress it because the answer is implausible. Could you clarify why this result indicates the presence of a direct pointer, rather than the target token being carried over as a side effect of other retrieval processes already in progress?

Causal Model
- When training the causal model variants that exclude one mechanism, are the weights of the remaining mechanisms retrained independently, or are they frozen from the full model? Clarifying this would help interpret the ablation results.

---

> ### Author Response · Authors · 2025-11-21
> **Rebuttal - Part I**
>
> We appreciate the reviewer’s favorable review and thoughtful suggestions, and are pleased they found our paper clear, and our methodology rigorous.
>
> **W1: Figures**
> We thank the reviewer for raising this point. We addressed this by adding clarifications to the captions of Figures 1,2, and 3\. We also now specifically explain what constitutes a mixed effect in the caption of Figure 2, connecting to its definition in Section 3.3. Explaining here as well for clarity: the mixed effects are cases not explained by any of the mechanisms. Analyses in Section 3.3, Figures 3 and 10, and appendix Section B, show that these are predictions distributed near the positional index. We show in these analyses that this occurs due to the fuzzy nature of the positional index.
>
> We hope that this addresses the reviewer’s concerns, and are happy to address any other clarity issues the reviewer would like to raise.
>
> **W2: Reflexive Mechanism**
> We appreciate the reviewer’s excellent point regarding the evidence for the existence of the reflexive mechanism, as well as the possible existence of a confounding suppressive mechanism. We have tested these confounds in our original submission in appendix Sections B.7 and C, however we fully agree that this evidence isn’t sufficiently present in the main text. We therefore move some of the relevant evaluations into the main text to address this.
>
> Overall, we prove the existence of the reflexive mechanism in two ways.
>
> First, in Section 3.4 we separate between patching a reflexive pointer and patching the answer entity. We do this by constructing a modified counterfactual dataset where the answer entity from the counterfactual prompt does not exist in the original prompt. Under this modified counterfactual, the model stops generating the counterfactual answer entity, instead falling back solely to the positional mechanism. This indicates that what was copied is a reflexive pointer that cannot be dereferenced. An alternative explanation raised by the reviewer is that the model might contain a mechanism that suppresses outputs corresponding to entities absent from the context. We exclude this possibility by running the same evaluation at layer $\\ell+1$, a point at which the model has already retrieved the correct answer. Here patching leads the model to output the counterfactual entity, showing that no such suppressive mechanism exists.
>
> Second, to further strengthen our claims, in Section F we block attention from the final token position to the target entity during the interchange intervention, for all layers $\\geq \\ell$. This leads to the same result as in the previous experiment, showing that the patched signal was a reflexive pointer that needed to be dereferenced, as opposed to the answer entity itself. Again, conducting the same experiment but patching at layer $\\ell+1$ and blocking attention for layers $\\geq \\ell+1$ leads to the model to output the counterfactual answer entity.
>
> Overall, these two experiments show that the model relies on a reflexive mechanism, where a direct pointer to the answer entity is used to retrieve it. If the reviewer has any more questions regarding clarity or conclusiveness, we would be happy to address them.

---

> > ### Author Response · Authors · 2025-11-21
> > **Rebuttal - Part II**
> >
> > **Q1: Sizes of the Entity Sets**
> > The sizes of the entity sets range from 23 (since that’s the minimum required for our interchange interventions with $n=20$) to 80\. We added this information to appendix Section A.1 which includes details about the binding tasks.
> >
> > **Q2: Figures**
> >
> > * We acknowledge that the labeling of the y-axis for the left plot is not sufficiently clear. The y-axis is the distribution of patch effects, while the `index=X` is meant to indicate that we are evaluating this distribution when patching a specific entity group index (0,10,19 respectively), evaluated for each layer in the model. In our revised version, we added a y-axis label encompassing all three rows, and clarified this in the caption.
> > * We address the clarity issue of the mixed label in our response to W1.
> > * In our experiments evaluating the next token prediction of the model, invalid entities are never predicted by the model and so are not shown or labeled in the results.
> >
> > **Q3: Reflexive Mechanism**
> > We address this alternative explanation for the reflexive mechanism experiment in our response to W2, where we detail our existing experiments which disprove this explanation, and move them to the main text for clarity.
> >
> > **Q4: Causal Model**
> > Our full causal model is composed of three sub-models, each of which is given its respective mechanism’s index ($i\_P,i\_L,i\_R$) and yields a logit distribution over the possible answers. When training the causal model variants that exclude one or more mechanisms, we omit entirely that mechanism from both the training process as well as the inference, and retrain each remaining mechanism’s weights from scratch. For example, for the $\\mathcal{M}\\setminus \\{L\_{\\text{one-hot}}\\}$ variant, our causal model is the one detailed in Equation 2 without the middle term representing the lexical mechanism. We added this clarification to the main text; if the reviewer feels that this should be further clarified we are happy to do so.
> >
> > We thank the reviewer for their time, and are happy to address any other issues that they may raise.

---

> > > ### Author Response · Authors · 2025-11-26
> > >
> > > Dear Reviewer zu1T,
> > >
> > > Thank you again for your favorable review and helpful comments. We have posted a detailed response addressing each of your comments and suggestions, and uploaded a revised version of our paper incorporating all of your feedback. Specifically, we added clarifications to the captions of several figures, as well as to Section 4 and the Appendix. Additionally, we moved several experiments validating the existence of the reflexive mechanism to the main text in Section 3.4. We believe these changes have significantly improved the quality of our paper.
> > >
> > > Thank you again for your time, and we are happy to address any remaining questions you may have.

---

### Meta-Review · Area_Chair_Xm4N · 2026-01-11

**Summary:**

This paper studies *in-context entity binding and retrieval* in transformer LMs (e.g., mapping “Ann loves pie” such that “Who loves pie?” retrieves “Ann”). Building on prior evidence for a **positional mechanism**, the paper argues that positional retrieval degrades as the number of entity groups in context grows (particularly for middle positions). The authors identify two additional retrieval mechanisms that supplement positional retrieval:

- **Lexical mechanism:** retrieval mediated by the queried bound counterpart (e.g., “pie” cues “Ann”).
- **Reflexive mechanism:** retrieval via a direct “pointer”-like signal to the target entity token.

The paper uses controlled original/counterfactual paired prompts and **interchange interventions (patching)** to causally separate these mechanisms, showing they can be set to disagree and that models mix them depending on conditions. Finally, the authors fit a simple causal/effective model combining the three mechanisms that reproduces next-token distributions with ~0.95 Jensen–Shannon similarity, and report robustness across nine models and ten binding tasks, including longer and more naturalistic/noisy contexts.

### Strengths
- **Clear, compelling causal methodology:** The counterfactual + interchange intervention setup is designed to isolate mechanism contributions beyond correlational probing, and reviewers found the interventional evidence persuasive.
- **Novel mechanistic characterization:** Identifying lexical and reflexive mechanisms extends the previously dominant positional account and provides a richer explanatory picture for failures of pure positional retrieval in longer contexts.
- **Breadth and robustness:** Results are replicated across multiple model families and tasks; additional robustness checks include filler text and (in revision) linguistic phrasing variability.

### Weaknesses
- **Mechanism distinctness framing:** One reviewer raised that lexical and reflexive may co-occur and asked for clearer justification that they are distinct causal variables rather than correlated manifestations of shared dynamics.
- **Template-heavy datasets:** Much of the analysis uses templated binding prompts; while the paper includes some robustness tests, the core setting is still relatively controlled.

**Reviewer Concerns:**

### Reviewer zu1T
- **Addressed:**
  - Figure clarity (axes/definitions of “mixed”): authors report caption and text clarifications.
  - Entity-set size range: explicitly answered and added to appendix.
  - Causal model ablation training details: clarified that ablated variants retrain remaining mechanism weights from scratch.
  - Reflexive mechanism ambiguity: authors moved additional validation into main text (new Section 3.4) and described controls intended to distinguish pointer patching vs answer-token carryover/suppression explanations.
- **Still outstanding (minor):**
  - Residual risk that reflexive “pointer” interpretation may remain subtle for some readers, but the revision appears to materially strengthen the evidential chain and presentation.

### Reviewer U696
- **Addressed:**
  - Interchange intervention implementation/localization: authors clarified layer selection and that interventions patch the last-token residual stream; reviewer follow-up indicates this resolved their main confusion, and authors additionally made explicit that this is **full vector patching**.
  - Reflexive mechanism explanation: added a figure and a dedicated section; reviewer states the figure is helpful and Section 3.4 helps.
- **Still outstanding (minor):**
  - Reviewer noted subtlety between query forms (“Who loves pie?” vs “What does Ann love?”) and suggested discussion; authors state they added discussion in Section 4. This seems largely addressed, though some subtle conceptual nuance may remain.

### Reviewer 4tWV
- **Addressed:**
  - Concern about “causal independence”: authors clarify they demonstrate causal controllability/independent effects at the mechanism level (via interventions where mechanisms are set to disagree and one is shifted while others are fixed).
  - Concern about templated prompts / linguistic variability: authors add a new evaluation (Appendix D.4) with phrasing variations and report consistency.
- **Still outstanding (minor):**
  - The reviewer’s stronger notion of independence at the level of *internal components/circuits* is not established (and authors explicitly defer that). Given the paper’s stated level of analysis (mechanism-level causal variables/effective model), this is not a blocker, but it remains an open direction.

**Reviewer Scores:**

All positive and I don't expect further change.

---

### Decision · Program_Chairs · 2026-01-26

Accept (Poster)